# Set1/COMPASS and Mediator are repurposed to promote epigenetic transcriptional memory

Agustina D'Urso[1], Yoh-hei Takahashi[2], Bin Xiong[3], Jessica Marone[1], Robert Coukos[1], Carlo Randise-Hinchliff[1], Ji-Ping Wang[3], Ali Shilatifard[2], Jason H Brickner[1]*

[1]Department of Molecular Biosciences, Northwestern University, Evanston, United States; [2]Department of Biochemistry and Molecular Genetics, Northwestern University, Chicago, United States; [3]Department of Statistics, Northwestern University, Evanston, United States

**Abstract** In yeast and humans, previous experiences can lead to *epigenetic transcriptional memory*: repressed genes that exhibit mitotically heritable changes in chromatin structure and promoter recruitment of poised RNA polymerase II preinitiation complex (RNAPII PIC), which enhances future reactivation. Here, we show that *INO1* memory in yeast is initiated by binding of the Sfl1 transcription factor to the *cis*-acting Memory Recruitment Sequence, targeting *INO1* to the nuclear periphery. Memory requires a remodeled form of the Set1/COMPASS methyltransferase lacking Spp1, which dimethylates histone H3 lysine 4 (H3K4me2). H3K4me2 recruits the SET3C complex, which plays an essential role in maintaining this mark. Finally, while active *INO1* is associated with Cdk8⁻ Mediator, during memory, Cdk8⁺ Mediator recruits poised RNAPII PIC lacking the Kin28 CTD kinase. Aspects of this mechanism are generalizable to yeast and conserved in human cells. Thus, COMPASS and Mediator are repurposed to promote epigenetic transcriptional poising by a highly conserved mechanism.

*For correspondence: j-brickner@northwestern.edu

## Introduction

Epigenetic transcriptional memory is a mitotically heritable form of 'priming' that changes the speed or strength of expression of select genes based on previous cellular experiences (*Brickner et al., 2007*; *D'Urso and Brickner, 2014*; *Light et al., 2010*; *2013*). A well-established model for transcriptional memory is the inducible inositol-1-phosphate synthase (*INO1*) gene in budding yeast (*Brickner et al., 2007*; *Light et al., 2010*, *2013*). Upon transcriptional activation, the *INO1* gene moves from the nucleoplasm to the nuclear periphery and physically interacts with the nuclear pore complex (NPC; *Ahmed et al., 2010*; *Brickner and Walter, 2004*). Upon repression, *INO1* remains associated with the NPC for up to four generations (*Brickner et al., 2007*; *Light et al., 2010*). Thus, maintenance of recently repressed *INO1* at the NPC represents an epigenetic state.

Active *INO1* and recently repressed *INO1* interact with different Nups through distinct and independent mechanisms. Each interaction involves different *cis*-acting *DNA zip codes*: DNA elements that are both necessary and sufficient for targeting to the nuclear periphery and interaction with the NPC (*Ahmed et al., 2010*; *Light et al., 2010*). Targeting of active *INO1* to the nuclear periphery requires two Gene Recruitment Sequences (GRSs) in the promoter that interact with the transcription factors Put3 and Cbf1 (*Brickner et al., 2012*; *Randise-Hinchliff et al., 2016a*). However, after repression, *INO1* remains associated with the periphery through a mechanism that is independent of the GRSs but requires a separate zip code, the Memory Recruitment Sequence (MRS) and the

**eLife digest** Cells respond to stressful conditions by changing which of their genes are switched on. Such stress-specific genes are typically switched off again when the conditions improve, but can remain primed and ready to be switched on again when needed. This phenomenon is known as "epigenetic transcriptional memory" and allows for a faster or stronger response to the same stress in the future. In fact, these memories can last for a long time, even after the cell divides many times.

Inside cells, most of the DNA is wrapped tightly around proteins called histones. To activate – or transcribe – a gene, the DNA must be re-packaged to allow better access for specific proteins including the enzyme called RNA polymerase II. This repackaging involves a number of changes including chemical modification of the histone proteins. Genes that have been previously transcribed under stress are packaged in a different way so that they are poised and ready for the next time they are needed. However, the details of this process were not clear.

Using yeast as a model, D'Urso et al. have dissected the changes that are responsible for priming genes to respond to future events. The yeast gene *INO1*, which shows transcriptional memory, was studied in cells by characterizing the proteins bound at and around the gene and the histone modifications in the region. D'Urso et al. found that a protein called Sfl1 bound to this gene only during transcriptional memory and that this binding was critical to start the phenomenon.

Further experiments showed that transcriptional memory also required altering two protein complexes that normally bind to genes when they are switched on. One complex, which includes an enzyme that modifies histones, was altered so that the histones at the *INO1* gene were marked in a unique way. The other complex was responsible for recruiting an inactive, poised form of RNA polymerase II to the gene, which allowed the gene to be activated when needed. In addition, D'Urso found that other genes that show transcriptional memory in yeast, as well as such genes in human cells, were also marked in the same ways.

A future challenge will be to understand how different conditions in different organisms can lead to transcriptional memory. Further studies could also explore how this memory phenomenon is inherited and how it influences an organism's fitness.

nuclear pore protein Nup100 (*Brickner et al., 2015*; *Light et al., 2010*). After repression, changes in chromatin structure (H2A.Z incorporation and dimethylation of H3K4; *Light et al., 2010*; *Santos-Rosa et al., 2002*) are required for *INO1* localization at the periphery (*Brickner et al., 2007*; *Light et al., 2010*) and binding of poised RNA polymerase II preinitiation complex (RNAPII PIC), which poises *INO1* for transcriptional reactivation (*Light et al., 2010*, *2013*). Loss of Nup100, H2A.Z or mutations in the MRS lead to loss of all aspects of *INO1* memory.

Epigenetic memory is a common phenomenon and may stem from an evolutionarily conserved mechanism. Upon repression, both *GAL1-10* (*Ng et al., 2003*; *Zhou and Zhou, 2011*) and *INO1* remain dimethylated on H3K4 (*Light et al., 2013*; *Santos-Rosa et al., 2002*). Furthermore, like *INO1*, *GAL1-10* localizes at the nuclear periphery and exhibits a faster rate of transcriptional reactivation for up to 8 generations (*Brickner et al., 2007*; *Tan-Wong et al., 2009*; *Kundu et al., 2007*; *Zacharioudakis et al., 2007*). Also, 77 of the genes induced by oxidative stress are activated more rapidly in yeast cells that have previously experienced salt stress, an effect that persists for four generations and requires the nuclear pore protein Nup42 (*Berry and Gasch, 2008*; *Gasch et al., 2000*; *Guan et al., 2012*). In *Arabidopsis*, heat shock alters the responsiveness of plants to heat stress for days and correlates with H3K4 methylation (*Ding et al., 2012*; *2013*; *Lämke et al., 2016*; *Liu et al., 2014*; *Sani et al., 2013*). In human cells, hundreds of the genes that are induced by IFN-γ exhibit stronger or faster induction in cells that have previously experienced IFN-γ (*Gialitakis et al., 2010*; *Light et al., 2013*). This effect persists through 4–7 cell divisions, is associated with H3K4 dimethylation and binding of poised RNAPII and requires physical interaction with the nuclear pore protein Nup98 (homologous to yeast Nup100; *Light et al., 2013*). Therefore, dimethylation of H3K4, binding of poised RNAPII and physical interaction with nuclear pore proteins may play a general, conserved role in memory to epigenetically enhance future gene expression. However, the potential

general role of H2A.Z in memory is unclear because loss of H2A.Z has pleiotropic effects (*Halley et al., 2010*).

Transcription in eukaryotes is commonly regulated through recruitment of RNAPII. However, transcription can also be stimulated through regulated elongation and, occasionally, through regulated initiation (*Kwak and Lis, 2013*; *Smith and Shilatifard, 2013*; *Spencer et al., 1990*). RNAPII often initiates transcription but after synthesizing 30–50 nt, becomes paused through the action of DSIF, NELF, GDOWN1 and the PAF complex (*Chen et al., 2015*; *Gilmour and Lis, 1986*; *Rougvie and Lis, 1988*; *Wada et al., 1998*; *Yamaguchi et al., 1999*; *Yu et al., 2015*). Paused RNAPII is phosphorylated on serine 5 of the carboxy terminal domain (CTD). Elongation is stimulated by recruitment of the super elongation complex, which phosphorylates serine 2 on the CTD (*Marshall et al., 1996*; *Ni et al., 2008*; *Smith and Shilatifard, 2013*). However, RNAPII recruitment does not always lead to initiation. A large number of inactive genes are bound by preinitiation RNAPII, suggesting that they are regulated upstream of transcription initiation in yeast and humans (*Kouzine et al., 2013*; *Radonjic et al., 2005*). Likewise, starvation of L1 larvae in *C. elegans* leads to RNAPII 'docking' over the promoters of ~750 genes involved in growth and development (*Maxwell et al., 2014*). This form of RNAPII appears to be neither active nor paused, suggesting that it has not initiated. Following previous expression, unphosphorylated, preinitiation RNAPII binds to the promoter of yeast and human genes that exhibit transcriptional memory, along with PIC components (*Light, 2010*, *2013*). Finally, the inducible Retinoic Acid Receptor β (*RARB*) promoter is bound to RNAPII and a partially assembled PIC in the absence of inducer (*Akoulitchev et al., 2000*; *Pavri et al., 2005*), suggesting that transcriptional poising may be a more general regulatory mechanism. Thus, all three steps of transcription can be regulated: PIC assembly/recruitment, transcription initiation and transcription elongation.

It is unclear how memory is initiated, how and why the memory-specific chromatin structure is established and how this leads to transcriptional poising. Using the yeast *INO1* gene as a model, we address these three questions, providing important new insight about the molecular mechanism of transcriptional memory. *INO1* memory is initiated by binding of the Sfl1 transcription factor to the MRS zip code specifically upon shifting from activating to repressing conditions. Sfl1 is necessary and sufficient to promote targeting to the nuclear periphery and is essential for all aspects of transcriptional memory. *INO1* memory is lost in strains that lack lysine 4 on histone H3 or upon conditional inactivation of the Set1/COMPASS complex, indicating that H3K4me2 is required for memory. During memory, the Set1/COMPASS histone methyltransferase is remodeled by dissociation of the Spp1 subunit. The resulting Spp1⁻ complex is capable of H3K4 dimethylation, but not trimethylation (*Miller et al., 2001*; *Morillon et al., 2005*; *Roguev et al., 2001*; *Schneider et al., 2005*; *Takahashi et al., 2009*; *Wood et al., 2003*). Set3, the eponymous member of the SET3C HDAC complex, is recruited by H3K4me2 to the *INO1* promoter through its PHD domain (*Kim and Buratowski, 2009*, *2013*; *Pijnappel et al., 2001*). Conditional inactivation of SET3C during memory rapidly disrupts both RNAPII binding and H3K4 dimethylation, suggesting that it is both the reader of this mark and is essential for its persistence. Finally, whereas a core Mediator subunit is required for RNAPII binding under both activating and memory conditions, the form of Mediator containing the Cdk8 kinase module (Cdk8⁺ Mediator) binds to the *INO1* promoter specifically during memory and is specifically required to recruit poised RNAPII and to enhance future expression.

To test the generality of our conclusions, we also probed the molecular mechanism of memory for stress-induced genes in yeast and IFNγ-induced genes in HeLa cells. During memory, these genes are marked with H3K4me2, bind RNAPII and are also associated with Cdk8 binding. Furthermore, salt stress-induced memory requires SET3C. This suggests that the mechanism of *INO1* memory is general and highly conserved.

## Results

### The Sfl1 transcription factor binds to the MRS to initiate *INO1* transcriptional memory

After repression, *INO1* remains associated with the NPC for up to four generations (*Brickner et al., 2007*; *Light et al., 2010*, *2013*). The interaction of recently repressed *INO1* with the NPC is controlled by the MRS zip code (5'-TCCTTCTTTCCC-3'; *Light et al., 2010*). Mutations in this DNA sequence

abolish all aspects of memory: after repression, *INO1* is not retained at the nuclear periphery, fails to incorporate H2A.Z or H3K4me2 into the promoter, does not retain RNAPII and exhibits a specific defect in the rate of reactivation (*Brickner et al., 2007*; *Light et al., 2010*; *2013*). Because the Sfl1 transcription factor binds to a sequence that is similar to the MRS (5'-TTCTTC-3') and shows genetic interactions with *NUP120* - a component of the Nup84 subcomplex that is required for transcriptional memory - we hypothesized that the Sfl1 transcription factor interacts with the MRS to promote memory (*Costanzo et al., 2010*; *Fujita et al., 1989*; *Light et al., 2010*; *Robertson and Fink, 1998*; *Zhu et al., 2009*). To test this hypothesis, we used chromatin immunoprecipitation (ChIP) against Sfl-GFP expressed in wild type and *mrs* mutant strains under activating (-inositol), repressing (+inositol) and memory conditions (-inositol → + inositol, 3 hr; *Figure 1A*). Sfl1 bound to the wild type *INO1* promoter specifically under memory conditions, but not to the *mrs* mutant *INO1* promoter (*Figure 1A*). Insertion of the MRS alone at the ectopic *URA3* locus leads to constitutive targeting to the nuclear periphery, suggesting that the factor(s) responsible for peripheral targeting is bound constitutively (*Light et al., 2010*). Indeed, at the ectopic MRS, Sfl1-GFP bound constitutively (*Figure 1—figure supplement 1A*). Furthermore, as previously observed, Sfl1-GFP bound to the *SUC2* promoter under all conditions (*Figure 1—figure supplement 1A*; *Song and Carson, 1998*). Finally, the levels and nuclear localization of Sfl1-GFP were not obviously different between these three conditions (*Figure 1—figure supplement 1B*). Thus, Sfl1 binds to the *INO1* promoter in an MRS-dependent and memory-specific manner that is apparently regulated in *cis* by its promoter context.

To test if Sfl1 mediates peripheral localization of *INO1* during memory, we used a chromatin localization assay (*Brickner et al., 2010*; *Egecioglu et al., 2014*). The *INO1* locus was tagged with an array of ~128 Lac repressor binding sites (LacO array) in cells expressing LacI-GFP (*Straight et al., 1996*; *Robinett et al., 1996*). Using confocal microscopy, the position of *INO1* was scored for colocalization with the nuclear envelope, visualized using a Pho88-mCherry fusion protein (*Figure 1B*). Because the shell constituting the outer 25–30% of the nuclear volume (*i.e.* the 200 nm closest to the nuclear envelope) is unresolvable from the nuclear envelope by light microscopy, a randomly localized spot within the nucleus is expected to colocalize with the nuclear envelope in ~30% of the cells (blue hatched line, *Figure 1B*; *Brickner and Walter, 2004*). In both the wild type and *sfl1Δ* strains, repressed *INO1* colocalized with the nuclear envelope in ~30% of the population and active *INO1* colocalized with the nuclear envelope in >50% of the cells under activating conditions (*Figure 1B*). Under memory conditions, *INO1* colocalized with the nuclear envelope in 52% of the wild type cells, but it colocalized with the nuclear periphery in only 32% of the *sfl1Δ* cells (*Figure 1B*). This phenotype is identical to that of the *mrs* mutant (*Light et al., 2010*) and suggests that Sfl1 is necessary for the maintenance of *INO1* at the nuclear periphery during memory.

To test if Sfl1 is sufficient to induce targeting to the nuclear periphery, we used a tethering strategy (*Randise-Hinchliff et al., 2016*). In a strain with a LacO array and LexA binding site integrated at *URA3* (a gene that normally localizes in the nucleoplasm; *Brickner and Walter, 2004*), we expressed either LexA or LexA-Sfl1. In the strain expressing LexA, *URA3* colocalized with the nuclear envelope in 32% of the population. However, in strains expressing Sfl1-LexA, *URA3* colocalized with the nuclear periphery in 53% of the population (*Figure 1B*). Therefore, Sfl1 is both necessary and sufficient to promote targeting to the nuclear periphery.

The output of transcriptional memory is RNAPII binding to the recently repressed promoter and faster induction in the future. Mutations that disrupt memory, such as the *mrs* mutation, lead to loss of RNAPII from the promoter during memory and a slower rate of reactivation, without affecting the rate of activation (*Light et al., 2010*). To test if Sfl1 is required for all aspects of memory, we performed ChIP against RNAPII during activation or reactivation and measured the association of RNAPII over the *INO1* promoter or coding sequence (*Figure 1C and D*). During activation, RNAPII recruitment to the *INO1* promoter and coding sequence was unaffected by loss of Sfl1 (*Figure 1C*) and the rate of *INO1* activation was unaffected (*Figure 1E*). However, during reactivation, RNAPII was not associated with the *INO1* promoter at t = 0 min in the *sfl1Δ* mutant strain (*Figure 1D*, left panel) and accumulated more slowing over the coding sequence (*Figure 1D*, right panel). This defect led to a specific reduction in the rate of *INO1* reactivation in the *sfl1Δ* mutant (*Figure 1F*), similar to the *mrs* mutant strain (*Light et al., 2010*). Thus, binding of the Sfl1 transcription factor to the *INO1* promoter upon repression promotes future *INO1* reactivation.

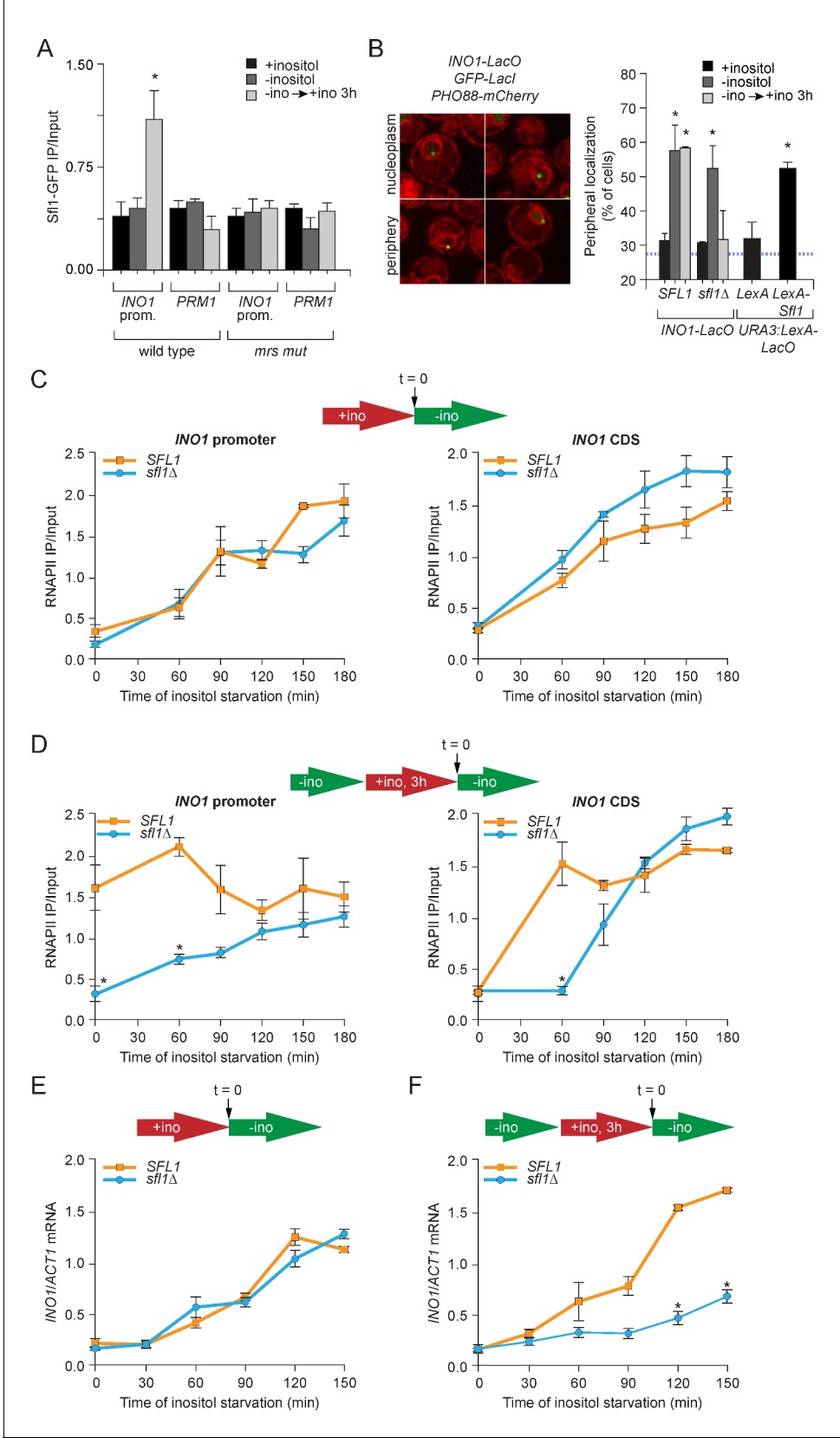

**Figure 1.** Sfl1 binds to the MRS to promote transcriptional memory. (**A**) Chromatin immunoprecipitation (ChIP) of Sfl1-GFP from wild type and *mrs* mutant *INO1* strains, quantified relative to the input fraction using primers to amplify the *INO1* promoter (−348 to −260) or the *PRM1* CDS, a repressed

*Figure 1 continued on next page*

*Figure 1 continued*

locus. The averages of three biological replicates are shown ± standard error of the mean. *p<0.05, compared with repressing conditions (Student's t-test). (B) Left: representative confocal micrographs of *INO1-LacO* in a strain expressing GFP-LacI and *PHO88-mCherry* scored as either nucleoplasmic or nuclear periphery. Right: quantified chromatin localization of the percentage of the population in which the indicated locus colocalized with the nuclear envelope. *INO1-LacO* in either a wild type or *sfl1Δ* strain was localized in cells grown in repressing (+inositol), activating (-inositol) or memory conditions (switched from medium lacking inositol to medium containing 100 μM inositol for 3 hr (−ino → +ino). *p<0.05, compared with repressing conditions (Student's t-test). *URA3:LexA-LacO* was localized in cells expressing either LexA or LexA-Sfl1 grown under repressing conditions. *p<0.05, compared with LexA alone (Student's t-test). The hatched blue line indicates the baseline for this assay (*Brickner and Walter, 2004*). (C and D) ChIP of RNA polymerase II from wild-type and *slf1Δ* cells fixed at indicated time points during activation (C) and reactivation (D). At time = 0, cells were shifted from repressing medium containing 100 μM inositol (red arrow in schematic) to medium without inositol (green arrow in schematic). For reactivation, cells were shifted from activating medium to repressing medium containing 100 μM inositol for 3 hr. Left panels were quantified relative to input using the *INO1* promoter primer set (-348 to -260, relative to the ATG); right panels were quantified relative to input using *INO1* coding sequence primer set (+663 to +798, relative to ATG). *p<0.05, compared with the repressing condition (Student's t-test). (E and F) *INO1* activation (E) or reactivation (F) in wild type and *sfl1Δ* cells (schematic as in C and D). Cells were harvested at the indicated time points, and *INO1* mRNA levels were quantified relative to *ACT1* mRNA levels by RT-qPCR. The averages of three biological replicates are shown ± standard error of the mean. *p<0.05, compared with the same time point in the *SFL1* strain (Student's t-test).

The following figure supplement is available for figure 1:

**Figure supplement 1.** Sfl1 binding to the *INO1* promoter is regulated by its context.

## H3K4me2 is an essential, memory-specific chromatin mark

Transcriptional memory is associated with histone modifications that are distinct from either the repressed or active states. Repressed *INO1* is hypoacetylated and unmethylated on H3K4 and active *INO1* is hyperacetylated and both di- and trimethylated on H3K4 (*Figure 2—figure supplement 1*; *Light et al., 2013*). However, upon repression, *INO1* loses histone acetylation and H3K4me3 (*Figure 2—figure supplement 1*; *Light et al., 2013*), but remains dimethylated on H3K4 (*Figure 2A*; *Light et al., 2013*). H3K4 dimethylation during memory occurs both over the *INO1* promoter and at the 5' end of the coding sequence (*Figure 2B*; *Santos-Rosa, 2002*; *Light et al., 2013*) and requires both Sfl1 and the MRS (*Figure 2A*). Therefore, Sfl1 is required for the persistent H3K4me2 associated with *INO1* memory.

Unlike H3K4me3, the H3K4me2 histone mark is associated with both active and inactive genes and functions to repress cryptic, non-promoter transcription (*Kim and Buratowski, 2009*; *Light et al., 2013*; *Margaritis et al., 2012*; *Pokholok et al., 2005*). H3K4me2 also correlates with poised promoters and epigenetic inheritance in yeast, plasmodium, plants, flies, worms, and humans (*Bevington et al., 2016*; *Gialitakis et al., 2010*; *Lämke et al., 2016*; *Light et al., 2013*; *Schaner et al., 2003*). To test the hypothesis that H3K4 methylation is necessary for *INO1* transcriptional memory, we used mutant yeast strains in which the sole copy of histone H3 has either an alanine (H3K4A) or an arginine (H3K4R) in place of lysine 4 (*Dai et al., 2008*). In the H3K4A and H3K4R mutant strains, RNAPII was recruited normally to the active *INO1* promoter, but was not recruited under memory conditions (*Figure 2C*). Therefore, Lysine 4 of histone H3 is necessary for *INO1* transcriptional poising.

Null mutations in the enzymes responsible for H3K4 methylation, Rad6, the H2B ubiquitin ligase or Set1, the catalytic subunit of the COMPASS histone H3 lysine 4 methyltransferase (*Briggs et al., 2001*; *Krogan et al., 2002*; *Roguev et al., 2001*), disrupt *INO1* peripheral localization and RNAPII binding during memory (*Light et al., 2013*). However, such null mutations cannot distinguish between H3K4 methylation during active transcription being a prerequisite for memory and H3K4 methylation being required for the establishment or perpetuation/inheritance of memory. Therefore, we employed the Anchor Away system (*Haruki et al., 2008*) to conditionally inactivate COMPASS to assess the importance of H3K4 methylation in the persistence/inheritance of memory. This system allows removal of a nuclear protein tagged with the FKBP12-rapamycin binding domain (FRB), expressed in a strain in which the ribosomal protein Rpl13A is fused to the FK506 binding protein (FKBP12; *Chen et al., 1995*; *Geisberg et al., 2014*; *Haruki et al., 2008*). Upon addition of rapamycin, the two will dimerize and, because ribosomes traffic through the nucleus during their biosynthesis (*Warner, 1999*), the nuclear protein will relocalize to the cytoplasm and be depleted from the nucleus. In the absence of the FRB fusion, the RPL13A-2xFKBP12 strain (HHY168) is resistant to

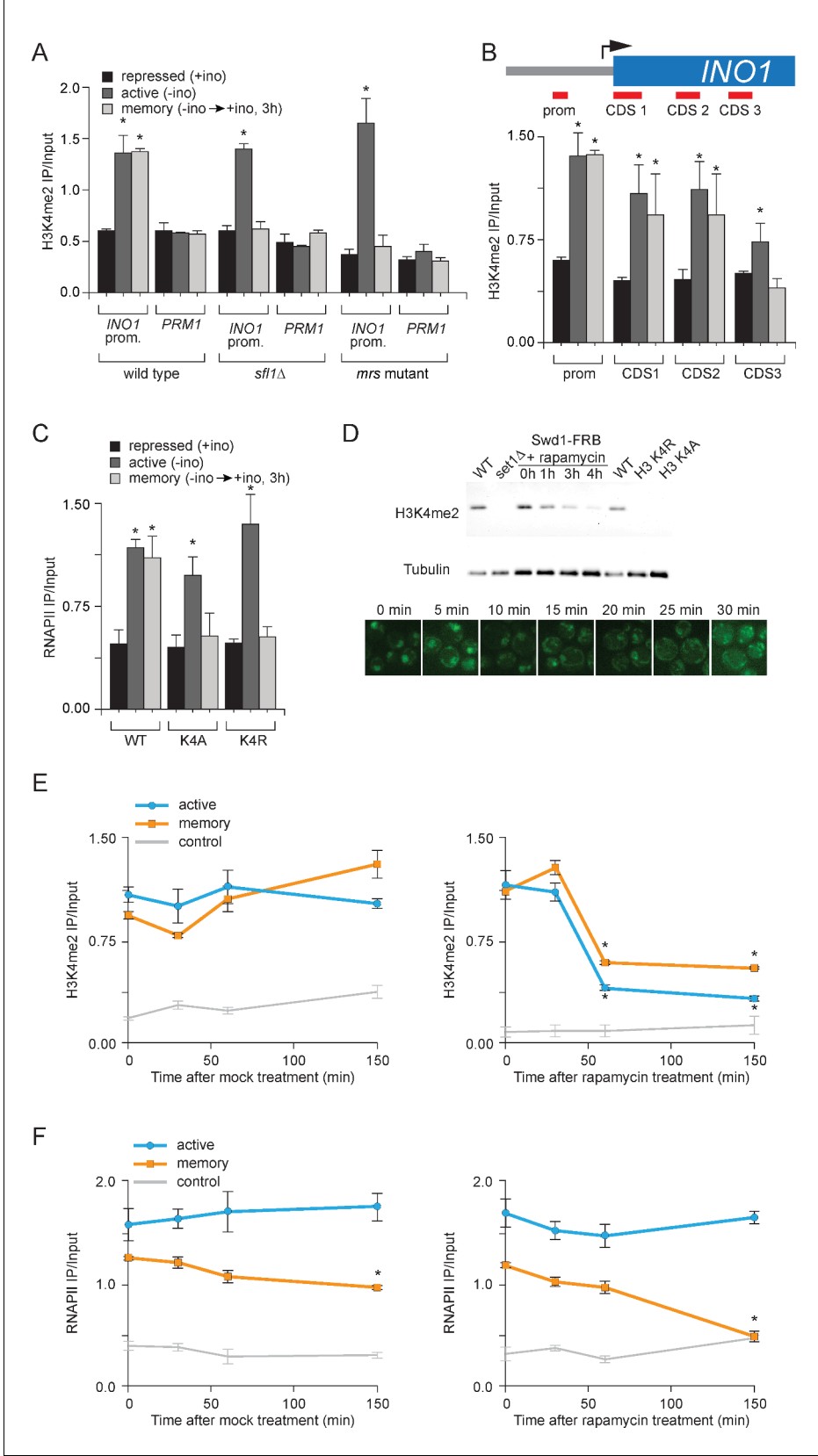

**Figure 2.** H3K4 dimethylation is an essential memory mark that is deposited by COMPASS. (**A** and **B**) Chromatin immunoprecipitation using anti-H3K4me2 from wild-type, *sfl1Δ* or *mrs* mutant strains grown under repressing, activating or memory conditions, quantified using the *INO1* promoter
*Figure 2 continued on next page*

*Figure 2 continued*

primer set (−348 to −260) or, as a negative control, the *PRM1* CDS primer set. *p<0.05, compared with the repressing condition (Student's t-test). (B) Recovery was quantified relative to input fractions using the promoter primer set or three different primer sets at the following postitions: pro, -348 to -260; CDS1, +41 to +161; CDS2, +361 to +499; CDS3, +663 to +798. (C) ChIP using anti-RNAPII from wild type and histone mutant (H3K4A or H3K4R) strains grown under repressing, activating and memory conditions using primers to the *INO1* promoter or *PRM1* CDS. *p<0.05, compared with the repressing condition (Student's t-test). (D) Top: immunoblot against H3K4me2 or Tubulin in whole cell extracts from the indicated strains. A strain expressing Rpl13-FKBP and having the COMPASS subunit Swd1 tagged with FRB-GFP was treated with 1 µg/ml rapamycin. Bottom: confocal micrographs of Swd1-FRB-GFP at the indicated times after addition of rapamycin. (E and F) ChIP of H3K4me2 (E) and RNAPII (F) from Swd1-FRB-GFP strain grown under activation (-ino) or memory conditions (−ino → +ino) using primers to amplify the *INO1* promoter or the *PRM1* CDS. Cells were fixed at the indicated times after addition of either DMSO (mock) or rapamycin. *p<0.05, compared with t = 0 (Student's t-test).

The following figure supplement is available for figure 2:

**Figure supplement 1.** Chromatin signature of transcriptional memory.

rapamycin (*Haruki et al., 2008*) and addition of rapamycin had no effect on H3K4 dimethyltion or RNAPII binding over the *INO1* promoter under either activating or memory conditions (*Figure 2—figure supplement 1*).

The Swd1 subunit of COMPASS was tagged with FRB-GFP. Within 30 min after adding rapamycin, Swd1-FRB-GFP relocalized from the nucleus to the cytoplasm but global levels of H3K4me2 dropped more slowly, with a half time of decay of ~2 hr (*Figure 2D*; *Soares et al., 2014*). Global H3K4me3 disappeared and H3K4me1 decreased over the same time period (*Figure 2—figure supplement 1F*). We monitored RNAPII binding and H3K4me2 over the *INO1* promoter under memory conditions (–inositol to +inositol for 3 hr) and activating conditions (-inositol) after addition of rapamycin. In mock-treated cells, H3K4me2 and RNAPII were maintained over the *INO1* promoter for ≥2.5 hr (*i.e.* 5.5 hr of repression), confirming that memory persists over this time period (*Figure 2E*). In cells treated with rapamycin, H3K4me2 was lost from *INO1* within 60 min of treatment in both activating and memory conditions (*Figure 2E*). Thus, COMPASS is required for the persistence of the histone mark during both active transcription and during memory.

Removal of COMPASS from the nucleus also led to loss of poised RNAPII from the *INO1* promoter specifically during memory. Rapamycin treatment resulted in a drop of RNAPII associated with the *INO1* promoter to baseline levels within 150 min under memory conditions, but had no effect on RNAPII association under activating conditions (*Figure 2F*). Therefore, loss of COMPASS-mediated H3K4 methylation specifically disrupted recruitment of poised RNAPII under memory conditions.

## COMPASS remodeling during transcriptional memory

Mono-, di- and trimethylated H3K4 show distinct genome-wide patterns and it is unclear how the COMPASS complex establishes and maintains such patterns (*Li et al., 2007*; *Shilatifard, 2006*). H3K4 trimethylation has been proposed to result from more stable association between the histone H3 tail and COMPASS (*Wood et al., 2007*). However, COMPASS lacking the Spp1 or Bre2 subunits is active for mono- and dimethylation of H3K4, but inactive for trimethylation of H3K4, suggesting that different methylation states could be due to remodeling of COMPASS (*Schneider et al., 2005*; *Soares et al., 2014*; *Takahashi et al., 2009*; *Thornton et al., 2014*). Also, the *spp1Δ* mutant has no effect on recruitment of RNAPII to promoters (*Morillon et al., 2005*). Transcriptional memory offers a unique system to explore this hypothesis.

To test if the composition of COMPASS changes during memory, we performed ChIP against GFP-tagged Swd1, Bre2, Sdc1 and Spp1 under repressing, activating and memory conditions. Under repressing conditions, none of the subunits of COMPASS associated with *INO1* and under activating conditions, all of the subunits associated with *INO1* (*Figure 3A*). However, under memory conditions, all of the subunits except Spp1 associated with *INO1* (*Figure 3A*). This suggested that COMPASS is remodeled during memory and that Spp1 is lost, producing an enzyme that is capable of H3K4 dimethylation, but not trimethylation. To confirm this observation, we monitored the association of Spp1 with the *INO1* promoter over time during the establishment of memory and during reactivation. Upon shifting cells from activating to repressing conditions, Spp1 was rapidly lost from the *INO1* promoter within 20 min (*Figure 3B*). Upon reactivation, Spp1 was recruited back to the

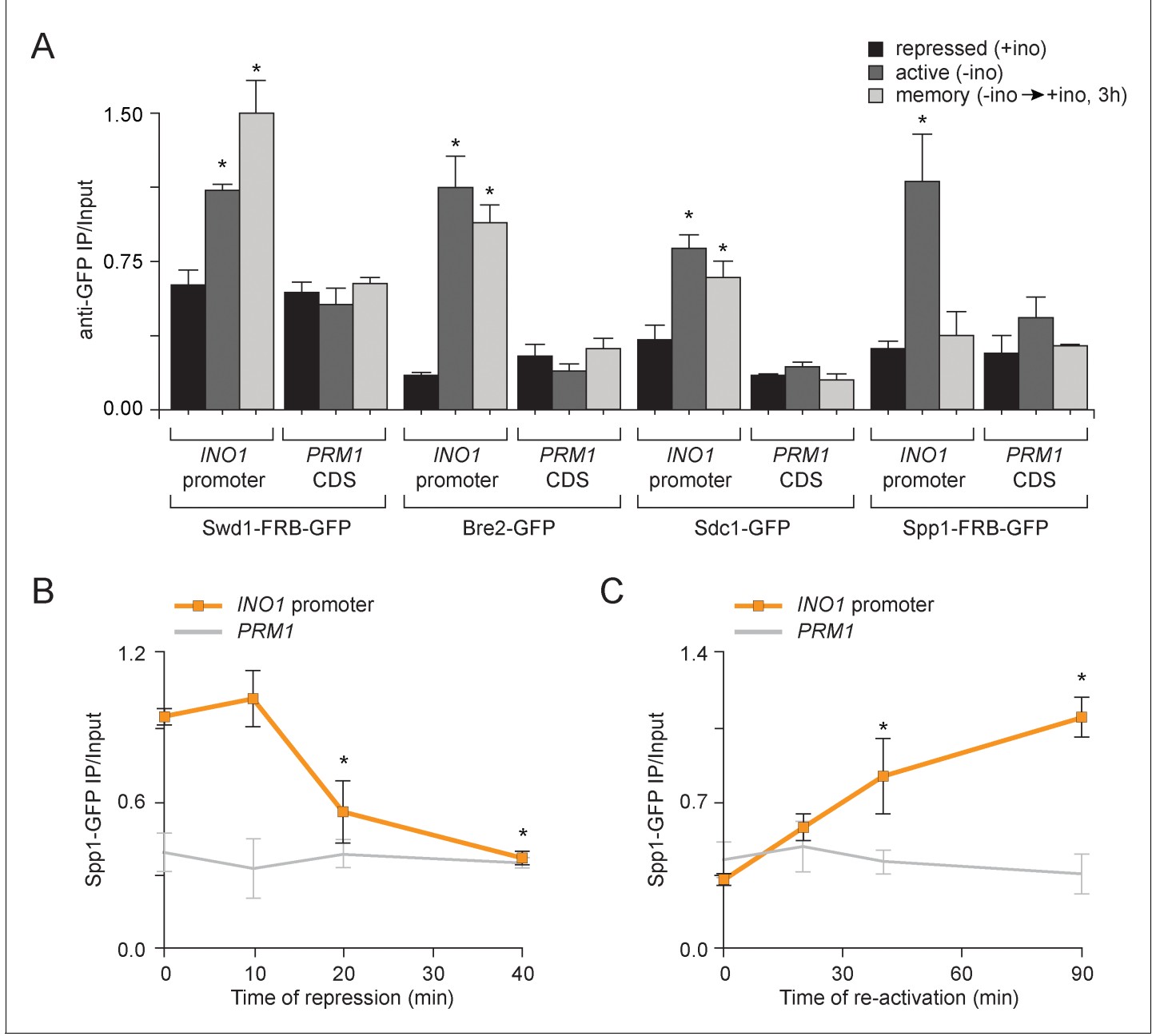

**Figure 3.** Transcriptional memory leads to remodeling of COMPASS. (**A**) ChIP against COMPASS subunits Swd1-GFP, Bre2-GFP, Sdc1-GFP, and Spp1-GFP from cells grown under repressing, activating or memory conditions. (**B** and **C**) ChIP against Spp1-GFP at the indicated times either after shifting cells from activating to repressing conditions (**B**) or after shifting cells back from repressing to activating conditions following 3 hr of repression (**C**). All ChIP experiments are averages of three biological replicates ± standard error of the mean, quantified relative to input using primers to amplify the *INO1* promoter (−348 to −260) or the *PRM1* CDS. *p<0.05, compared with the repressing condition (**A**) or compared with the 0 min time point (**B** and **C**) (Student's t-test).

*INO1* promoter within 30–45 min (*Figure 3C*). This suggests either that COMPASS is actively remodeled on the *INO1* promoter upon repression or that two distinct forms of COMPASS exist in vivo and that these forms are differentially recruited to catalyze trimethylation of H3K4 during activating conditions and dimethylation of H3K4 during memory

## SET3C binds H3K4me2 and promotes its persistence during memory

To understand the functional role of H3K4me2 in memory, we asked if the SET3C histone deacetylase complex recognizes this mark to promote memory. Set3 possesses a PHD domain that directly interacts with H3K4me2 (*Kim and Buratowski, 2009*; *Kim et al., 2012*) and *set3Δ* mutants disrupt *INO1* memory (*Light et al., 2013*). To confirm that Set3 is recruited to the *INO1* promoter by 'reading' the H3K4me2 mark, we performed ChIP against Set3-FRB-GFP. Indeed, Set3 binding to *INO1* reflects the dimethylation of H3K4; Set3 bound under both activating and memory conditions, but not under repressing conditions (*Figure 4A*). Binding of Set3 during memory was lost in cells lacking Sfl1, consistent with role of Sfl1 in promoting H3K4me2 specifically during memory (*Figure 4B*). Also, mutation of tryptophan 104 to alanine in the PHD domain of Set3, which blocks binding to

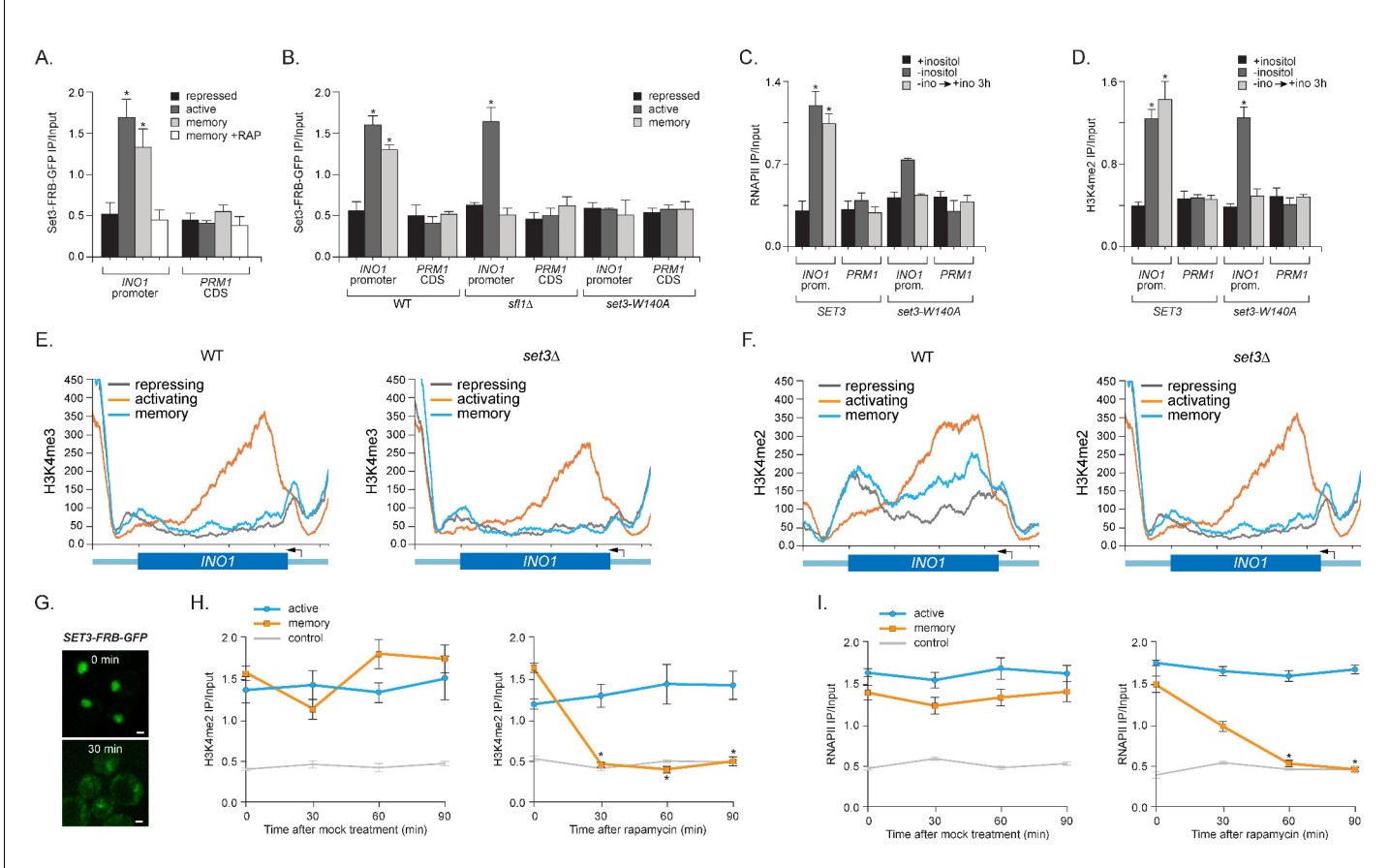

**Figure 4.** Set3 recruitment to the *INO1* promoter under memory conditions requires both Sfl1 and the PHD finger. (A) ChIP against Set3-GFP from cells grown under repressing, activating or memory conditions +/- rapamycin. (B) ChIP against SET3-GFP from wild type, *sfl1Δ* or *set3-W140A* cells grown under repressing, activating or memory conditions. (C and D) ChIP against RNAPII (C) and H3K4me2 (D) from wild type an *set3-W140A* strains grown under repressing, activating or memory conditions. For A–D, *p<0.05, compared with the repressing condition (Student's t-test). (E and F) ChIP sequencing against H3K4me3 (E) and H3K4me2 (F) from wild type (left) and set3Δ (right) strains grown under repressing, activating and memory conditions using primers to amplify the *INO1* promoter (−348 to −260) or the *PRM1* CDS. (G) Confocal micrographs of Set3-FRB-GFP at the indicated times after addition of rapamycin. (H and I) ChIP of H3K4me2 (H) and RNAPII (I) from Set3-FRB-GFP strain grown under activation (-ino) or memory conditions (−ino → +ino). Cells were fixed at the indicated times after addition of either DMSO (mock) or rapamycin. All ChIP experiments were quantified by qPCR and are plotted as averages of three biological replicates ± standard error of the mean. *p<0.05, compared with t=0 (Student's t-test).

The following source data and figure supplement are available for figure 4:

**Source data 1.** Genome wide analysis in wild type and *set3Δ* cells for H3K4me2 and H3K4me3 Chip-Seq.

**Figure supplement 1.** Loss of Set3 has no effect on histone acetylation or H3K4me3 at the *INO1* promoter.

H3K4me2 (*Kim and Buratowski, 2009*; *Pijnappel et al., 2001*), disrupted Set3 binding under all conditions (*Figure 4B*). This mutation also disrupted RNAPII binding during memory (*Figure 4C*). Thus, Set3C is recruited under both activating and memory conditions to the *INO1* promoter by recognition of H3K4me2 and this interaction is required for RNAPII binding under memory conditions.

Set3 is also required for persistent H3K4 dimethylation during memory. Substitution of alanine for tryptophan 140 in the Set3 PHD finger (or loss of Set3; not shown) resulted in loss of H3K4me2 on the *INO1* promoter under memory conditions, but had no effect on the H3K4me2 over *INO1* under activating conditions (*Figure 4D*). ChIP-seq against H3K4me3 and H3K4me2 in wild type and *set3Δ* strains revealed that *INO1* memory is associated with a Set3-dependent H3K4 dimethylation over the 5' end of the gene (*Figure 4E and F*; complete dataset: doi:10.5061/dryad.93fv2). The pattern of H3K4 dimethylation observed by ChIP-seq was similar to that revealed by ChIP qPCR (*Figure 2B*), although the signal over the promoter was less apparent. This may be due to differences in shearing efficiency, aspects of library preparation or differences in normalization between the two techniques. Regardless, we observed clear Set3-dependent H3K4me2 over the *INO1* gene during memory. Trimethylation of H3K4 was observed only under activating conditions and was not Set3 dependent (*Figure 4E*; *Figure 4—figure supplement 1A*). Furthermore, loss of Set3 resulted in neither trimethylation of H3K4 nor hyperacetylation of H3/H4 over the *INO1* promoter during memory (*Figure 4E*; *Figure 4—figure supplement 1*). These results suggest that recognition of H3K4me2 by the Set3 PHD domain is required for the persistence of this mark during memory, but is dispensable for the deposition of H3K4me2 under activating conditions. Although Set3 associates with two different histone deacetylases, loss of Set3 does not alter the acetylation of histones over the *INO1* promoter under any of these conditions (*Figure 4—figure supplement 1*).

Set3 null mutants have both positive and negative effects on transcription (*Kim et al., 2012*; *Wang et al., 2002*). *SET3* knockout mutants or null mutants having a loss-of-function mutation in the PHD domain also showed a defect in the expression of *INO1* and lower RNAPII binding under activating conditions (*Figure 4C* and data not shown). To confirm that the role of Set3 is direct and specific, we utilized the Anchor-Away system. Removal of Set3-FRB-GFP from the nucleus upon addition of rapamycin led to rapid loss of both H3K4 dimethylation and RNAPII binding under memory conditions but not under activating conditions (*Figure 4G–I*). Therefore, Set3 is required for the persistence of H3K4me2 and RNAPII recruitment under memory conditions. As was observed with inactivation of COMPASS, loss of H3K4me2 preceded loss of RNAPII. This suggests that Set3 has a direct and continuous role in the perpetuation of transcriptional memory by both recognizing H3K4me2 and maintaining this mark after repression.

## Molecular requirements for PIC assembly during memory

The binding of RNAPII during memory is not simply a residual effect of previous transcription or the slow disassembly of the PIC after repression. The binding of RNAPII during memory requires the MRS, H2A.Z, Sfl1, COMPASS, SET3C and Nup100; loss of these factors leads to rapid loss of RNAPII from the promoter specifically during memory (*Figures 1,2,4*; *Light et al., 2010*; *2013*). Furthermore, RNAPII association is *epigenetic*, persisting for ≥6 hr (3–4 generations) after repression, suggesting that it is bound to the promoter of the gene that had been expressed as well as the promoter of that gene in the daughters, granddaughters and great-granddaughters of that cell (*Light et al., 2010*). Finally, *INO1* transcriptional memory is also associated with binding of components of the preinitiation complex, including TBP, TFIID, TFIIA, TFIIB, TFIIE, TFIIF and TFIIH (*Light et al., 2013*). Thus, the most parsimonious interpretation of these results is that memory recruits the PIC, leading to a poised form of the promoter that affects future activation rates.

To explore the molecular requirements for PIC assembly during memory and compare them to the requirements for PIC assembly during active transcription, we used the Anchor-Away system. Removal of TBP (Spt15), Mediator (Med1), TFIIH (Tfb1) and TFIIK (Kin28) from the nucleus was monitored by confocal microscopy and the effect of loss of each of these components was measured by ChIP against RNAPII and H3K4me2. Inactivation of Spt15, Med1 and Tfb1 caused RNAPII levels to drop over the *INO1* promoter under both activating and memory conditions, suggesting that TBP, Mediator and TFIIH are necessary for recruitment and stabilization of both active and poised PIC (*Figure 5B, E, H and K*). However, depletion of Kin28 (Cdk7 in mammals), the kinase module of TFIIK that phosphorylates the CTD of RNAPII upon initiation, disrupted RNAPII binding to the *INO1* promoter under activating conditions, but not under memory conditions (*Figure 5K*). This is

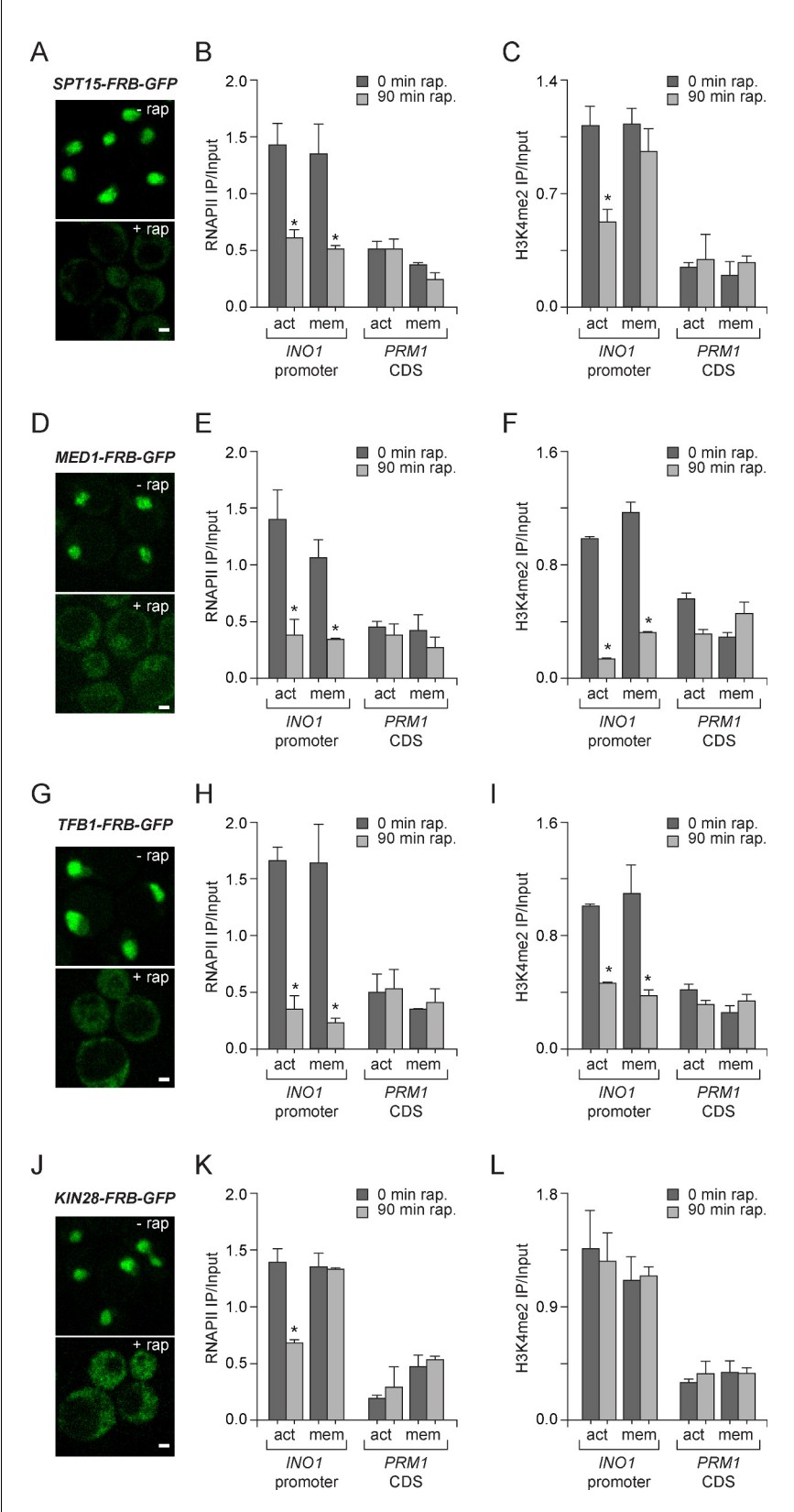

**Figure 5.** Molecular requirements for PIC assembly during transcriptional memory. (**A**, **D**, **G** and **J**) Confocal micrographs of the indicated proteins fused to FRB-GFP before or after treatment with rapamycin for 90 min. (**B**, **E**, **H** and **K**) ChIP against RNAPII from strains expressing Spt15-FRB-GFP (**B**),

*Figure 5 continued on next page*

*Figure 5 continued*

Med1-FRB-GFP (E), TFB1-FRB-GFP (H) or Kin28-FRB-GFP (K), grown under either activating or memory conditions, before or after treatment 1 µg/ml of rapamycin. (C, F, I and L) ChIP against H3K4me2 from strains expressing Spt15-FRB-GFP (C), Med1-FRB-GFP (F), TFB1-FRB-GFP (I) or Kin28-FRB-GFP (L), grown under either activating or memory conditions, before or after treatment 1 µg/ml of rapamycin. All ChIP experiments are averages of three biological replicates ± standard error of the mean, quantified as in panel 1A, using primers to amplify the *INO1* promoter (−348 to −260) or the *PRM1* CDS. Mock treatment had no effect (not shown). *p<0.05, compared with 0 min rapamycin (Student's t-test).

consistent with the observation that Kin28 is not bound to the *INO1* promoter under memory conditions and that the RNAPII bound during memory is unphosphorylated on the CTD (*Light et al., 2013*). Kin28 phosphorylation of the CTD regulates the interaction of Mediator with PIC, permitting promoter escape, which does not occur during memory (*Jeronimo and Robert, 2014*; *Wong et al., 2014*). This suggests that PIC assembly during memory proceeds through a mechanism very similar to PIC assembly during active transcription, but is arrested upstream of Kin28/TFIIK recruitment, which maintains the PIC in a poised state.

H3K4me2 associated with active *INO1* was also lost in cells in which Spt15, Med1 and Tfb1 were removed from the nucleus (*Figure 5C,F and I*). In contrast, H3K4me2 associated with *INO1* during memory was lost in cells in which Med1 and Tfb1 were removed from the nucleus but was unaffected by removal of Spt15 from the nucleus (*Figure 5C,F and I*). Curiously, removal of Kin28 from the nucleus by treatment with rapamycin for 90 min had no effect on H3K4me2 under any conditions (*Figure 5L*). This suggests RNAPII binding can be disrupted without affecting H3K4me2 and supports the notion that PIC assembly during memory is downstream of histone modification.

## Cdk8[+] Mediator plays a conserved role in transcriptional memory

The effect of inactivation of Med1 on RNAPII recruitment during memory was unexpected because ChIP against the Gal11 subunit of Mediator had previously suggested that Mediator was absent during memory (*Light et al., 2013*). To confirm that Mediator binds under memory conditions, we performed ChIP against GFP-tagged Med1 (*Figure 6A*). Med1-GFP bound to the *INO1* promoter both under activating and memory conditions, suggesting that Mediator is present under both (*Figure 6A*).

Because the ChIP results suggested that certain Mediator subunits were lost during memory and others were retained, it seemed plausible that the *INO1* promoter might interact with two different forms of Mediator under activating and memory conditions. In particular, we hypothesized that memory might involve the Cdk8[+] form of Mediator because Sfl1 and the NPC-associated TREX-2 complex associate with the Cdk8[+] Mediator (*Schneider et al., 2015*; *Song and Carlson, 1998*) and regulation of the poised *RARB* gene involves Cdk8[+] Mediator, which dissociates upon activation (*Pavri et al., 2005*). Consistent with a memory-specific role of Cdk8+ Mediator, components of the Cdk8 module of Mediator (Med13 and Ssn8; *Tsai et al., 2014*), showed binding to the *INO1* promoter only under memory conditions (*Figure 6A*). Likewise, ChIP against Ssn3-FRB-GFP, the yeast Cdk8 homolog protein, revealed that that it is also bound to *INO1* only during memory (*Figure 6B*). Overall, this suggests that the Cdk8[+] Mediator is recruited, potentially by Sfl1, to the *INO1* promoter during memory.

To assess the functional significance of Ssn3 binding to the *INO1* promoter, we conditionally removed Ssn3-FRB-GFP from the nucleus using the Anchor-Away system (*Figure 6C*). Conditional inactivation was critical because *ssn3Δ* mutants show global derepression of many genes, including *INO1* (data not shown; *Hampsey, 1998*; *van de Peppel et al., 2005*). Rapamycin was added either under repressing conditions or 2 hr after *INO1* memory was established. Conditional inactivation of Ssn3 did not lead to RNAPII binding under repressing conditions, suggesting that the phenotype of the null mutant is not observed immediately upon inactivation of Ssn3 (*Figure 6C*, right panel). In contrast, addition of rapamycin under memory conditions led to loss of RNAPII from the *INO1* promoter within 60 min (*Figure 6C*, right panel). This result suggests that Cdk8[+] Mediator is required to recruit poised PIC during memory.

To determine if inactivation of Ssn3 disrupts transcriptional memory, we also measured *INO1* activation and reactivation rates. Cells grown under either repressing or memory conditions were treated with rapamycin for 45 min before switching them to activating/reactivating conditions and *INO1* mRNA levels were measured over time. The rate of *INO1* activation was unaffected by removal of Ssn3

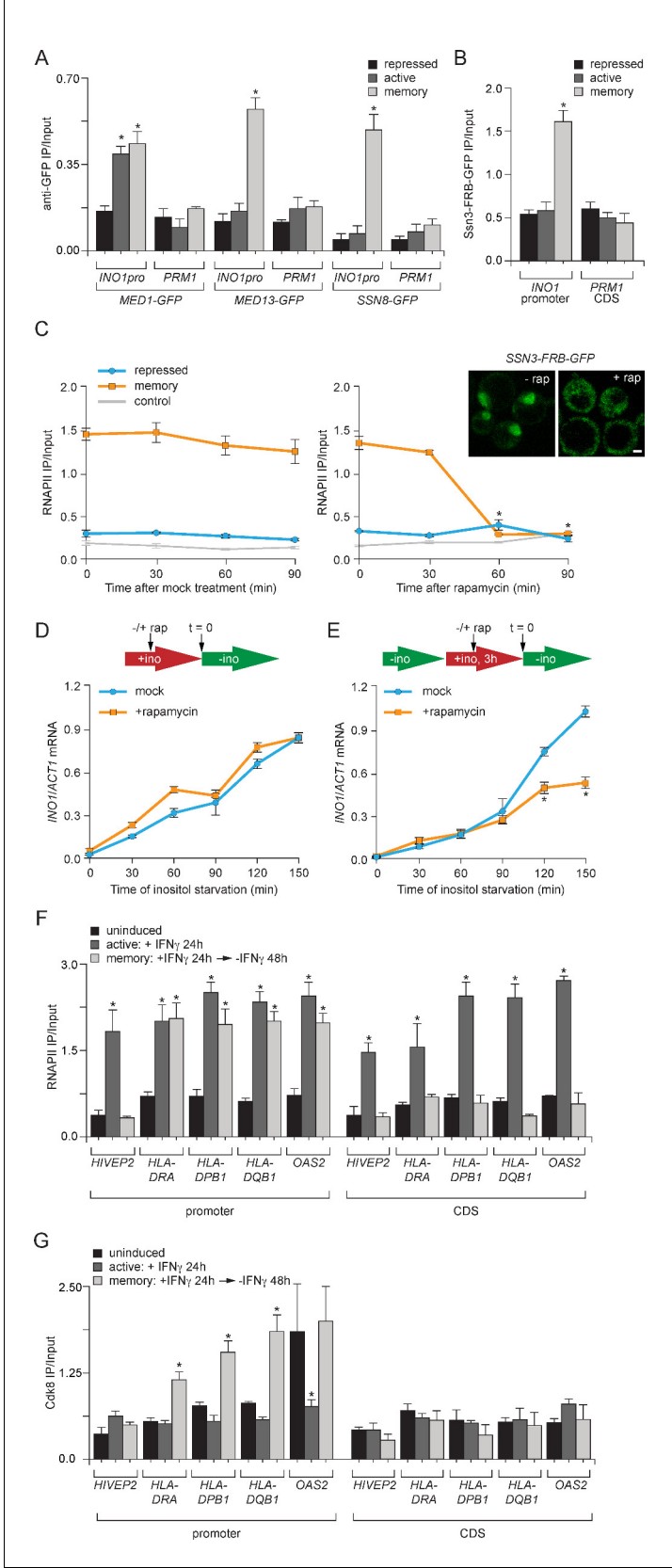

**Figure 6.** Transcriptional memory leads to Ssn3/Cdk8-dependent poised preinitiation complex. (**A**) ChIP against Med1-GFP, Med13-GFP or Ssn8-GFP from cells grown under repressing, activating or memory conditions. (**B**) ChIP against Ssn3-FRB-GFP from cells grown in repressing, activating or

*Figure 6 continued on next page*

*Figure 6 continued*

memory conditions. (**A** and **B**) *p<0.05, compared with the repressing condition (Student's t-test). (**C**) ChIP against RNAPII from strains expressing Ssn3-FRB-GFP grown under either repressing or memory conditions, before or after treatment 1 µg/ml of rapamycin using primers to amplify the *INO1* promoter (−348 to −260) or the *PRM1* CDS. Inset: confocal micrographs of Ssn3-FRB-GFP expressing cells before or after treatment with 1 mg/ml of rapamycin for 30 min. *p<0.05, compared with t = 0 (Student's t-test). (**D** and **E**) *INO1* activation (**D**) or reactivation (**E**) in Ssn3-FRB-GFP cells. For activation at time = 0, cells were shifted from medium containing 100 µM inositol (repressing conditions; red arrow in schematic) to medium without inositol (activating conditions; green arrow in schematic). For reactivation, cells were shifted from activating medium to repressing medium containing 100 µM inositol for 3 hr. Cells were treated ±1 µg/ml rapamycin for 45 min before transferring to activating conditions. Cells were harvested at the indicated time points, and *INO1* mRNA levels were quantified relative to *ACT1* mRNA levels by RT-qPCR. The averages of three biological replicates are shown ± standard error of the mean. *p<0.05, compared with the same time point in the mock-treated culture (Student's t-test). (**F** and **G**) ChIP against RNAPII (**F**) or Cdk8 (**G**) from HeLa cells before, during (24 hr) or 48 hr after treatment with 50 ng/mL Interferon-γ. Recovery of the indicated promoters or coding sequences (CDS) of genes that exhibit transcriptional memory (*HLA-DRA*, *HLA-DPB1*, *HLA-DQB1* and *OAS2*) and a gene that does not (*HIVEP2*) was quantified relative to input by qPCR. *p<0.05, compared with the uninducing condition (Student's t-test). (**A-F**) Averages of three biological replicates ± standard error of the mean.

from the nucleus (*Figure 6D*). However, the rate of *INO1* reactivation was slower in cells that had been treated with rapamycin (*Figure 6E*). This suggests that Cdk8⁺ Mediator plays an essential and specific role in transcriptional poising.

To test if the role played by Cdk8⁺ Mediator in memory is conserved in mammals, we asked if genes that exhibit IFN-γ memory in HeLa cells also associate with Cdk8⁺ Mediator. We performed ChIP against RNAPII and Cdk8 from untreated cells (uninduced), after 24 hr treatment with IFN-γ (active) or 48 hr after removal of IFN-γ (memory). The promoters of genes that show IFN-γ memory are marked by H3K4 dimethylation and bind to poised RNAPII following treatment with IFN-γ (*Figure 6F*; *Gialitakis et al., 2010*; *Light et al., 2013*). As a control, we monitored the promoter of a gene that is IFN-γ inducible but does not show memory (*i.e. HIVEP2*), which showed RNAPII bound only in the presence of IFN-γ (*Figure 6F*; *Gialitakis et al., 2010*; *Light et al., 2013*). Cdk8 did not associate with the *HIVEP2* promoter under any condition (*Figure 6G*). In contrast, after removal of IFN-γ, Cdk8 associated with the promoters of all four genes that exhibit transcriptional memory: *HLA-DRA*, *HLA-DPB1*, *HLA-DQB1* and *OAS2* (*Figure 6G*). For *HLA-DRA*, *HLA-DPB1* and *HLA-DQB1*, this was specific; Cdk8 binding was only observed after removal of IFN-γ (*Figure 6G*). At the *OAS2* promoter, Cdk8 was bound both prior to treatment with IFN-γ and under memory conditions (*Figure 6G*). This suggests that Cdk8⁺ Mediator binding is a highly conserved feature of transcriptional memory.

## Molecular mechanism of salt stress-induced epigenetic transcriptional memory

Comprehensively defining the scope of transcriptional memory is challenging because any stimulus that regulates transcription could lead to memory for a subset of induced genes. For example, ~ 260 of the ~600 genes that are upregulated by IFN-γ show faster or stronger expression in cells that have been previously exposed to IFN-γ (*Light et al., 2013*). ChIP-seq against H3K4me3 and H3K4me2 in wild type and *set3Δ* strains under repressing, activating or memory conditions for *INO1* identified many loci whose dimethylation is Set3-dependent, loci that are induced by inositol starvation but do not show memory, genes whose expression is Set3-dependent and genes that show Set3-dependent H3K4me2 under all conditions (*Figure 4—source data 1*). However, only a few of these loci correspond to genes that are co-regulated with *INO1* and show Set3-dependent H3K4me2 under memory conditions (*Figure 4—source data 1*). This suggests that *INO1* memory is remarkably specific. Therefore, to test the generality of the molecular mechanism of *INO1* transcriptional memory, we focused on yeast genes that show salt stress memory (*Guan et al., 2012*). Previous exposure to 0.7M sodium chloride leads to faster or stronger induction of ~75 $H_2O_2$-inducible genes for up to 4 generations and this requires the nuclear pore protein Nup42 (*Guan et al., 2012*). We confirmed this observation for several of these genes (*PGM2*, *PMT5* and *YGP1*, *Figure 7A*; and *USV1*, not shown; *Guan et al., 2012*). *HSP31* is $H_2O_2$-inducible, but does not exhibit memory and is induced identically in cells that have previously experienced high salt and cells that have not (*Figure 7A*; *Guan et al., 2012*). ChIP revealed that the promoters of genes that exhibit memory were associated with RNAPII and H3K4me2 after treatment with salt, but not without treatment with

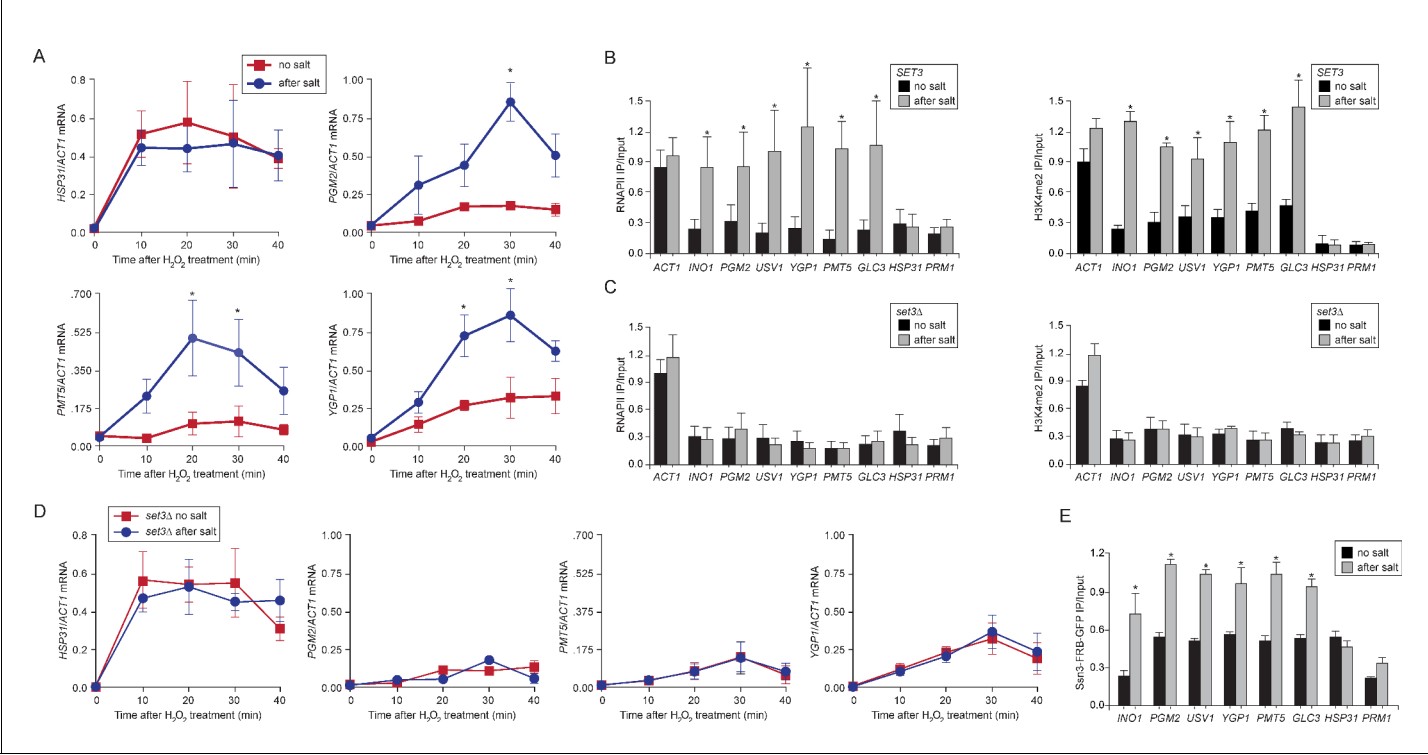

**Figure 7.** Salt-induced transcriptional memory leads to dimethylation of H3K4 and binding of poised RNAPII. (**A**) mRNA levels of three genes that exhibit transcriptional memory (*PGM2, PMT5 & YGP1*) and one gene that does not (*HSP31*) at the indicated times after treatment with 0.5mM $H_2O_2$. Prior to treatment with $H_2O_2$, cells were grown either in rich media (no salt; red lines) or treated with 0.7M NaCl for 1 hr and then allowed to recover for 2 hr in rich media (after salt; blue lines). mRNA levels were quantified relative to *ACT1* by RT-qPCR. Shown are the averages of three biological replicates ± standard error of the mean. *$p < 0.05$, compared with the same time point in the no salt culture (Student's t-test). (**B**) mRNA levels of three genes that exhibit transcriptional memory (*PGM2, PMT5 & YGP1*) and one gene that does not (*HSP31*) from *set3Δ* mutant cells at the indicated times after treatment with 0.5 mM $H_2O_2$ same data as in (**A**). (**C** and **D**) ChIP against RNAPII (**C**), H3K4me2 (**D**) from wild-type and *set3Δ* cells grown either in the absence of salt (no salt) or treated with 0.7M NaCl for 1 hr and allowed to recover for 2 hr in rich medium (after salt). (**E**) ChIP against Ssn3-FRB-GFP cells grown either in the absence of salt (no salt) or treated with 0.7M NaCl for 1 hr and allowed to recover for 2 hr in rich medium (after salt). All ChIP experiments are averages of three biological replicates ± standard error of the mean, quantified as in panel 1A, using primers to amplify the promoters of the indicated genes. *$p < 0.05$, compared with the no salt condition (Student's t-test).

salt (*Figure 7B*). *HSP31* did not show this pattern (*Figure 7B*). Finally, we discovered that salt treatment also induced *INO1* memory (*Figure 7B*), suggesting some overlap in the regulation of these two forms of memory. Thus, salt-induced memory leads to H3K4 dimethylation and RNAPII binding.

Loss of Set3 disrupted salt-induced memory. In strains lacking Set3, RNAPII binding and H3K4 dimethylation were lost (*Figure 7C*) and the reactivation of these genes after salt treatment was reduced (*Figure 7D*). This effect is specific for genes that exhibit salt stress-induced memory: the rate of activation of *HSP31* was unaffected by loss of Set3. Salt-induced memory was not dependent on Sfl1; loss of Sfl1 had no effect on RNAPII binding and H3K4 dimethylation after salt exposure (data not shown). Therefore, salt-induced memory is Set3-dependent but Sfl1-independent, suggesting that Set3 represents a general requirement downstream of H3K4me2 for transcriptional memory.

Finally, we tested if Cdk8[+] Mediator bound to the promoters of genes that show salt-induced memory. Like *INO1*, Ssn3 bound to the promoters of these genes in cells that had been previously treated with salt (*Figure 7E*). Among all of the stress-responsive genes, including *HSP31*, we observed a higher background of Ssn3 binding in the untreated cells. However, this binding increased during memory, suggesting that Cdk8[+] Mediator plays a general role in promoting epigenetic transcriptional poising, albeit via different transcription factor-dependent recruitment mechanisms.

# Discussion

As a model for the general phenomenon of environmentally-induced epigenetic transcriptional memory, we have defined the molecular basis of yeast *INO1* memory. *INO1* memory requires a *cis*-acting MRS element that functions as a DNA zip code to target *INO1* to the nuclear periphery after repression (*Light et al., 2010*). We propose that memory is initiated by the regulated binding of the Sfl1 transcription factor. Sfl1 is both necessary and sufficient to promote targeting to the nuclear periphery and loss of Sfl1 disrupts all aspects of *INO1* memory. If Sfl1 binding initiates memory, then the duration and heritability of memory may reflect the regulation of Sfl1 binding to the MRS. Sfl1 binds to the *INO1* promoter in an MRS-dependent manner specifically during memory. Because Sfl1 binding to other sites is constitutive, it seems that the regulation of Sfl1 binding is context-dependent. Sfl1 binding to the MRS may be influenced in *cis* by other transcription factors or by the changes in chromatin structure that are associated with transcriptional memory. Understanding the regulation of Sfl1 binding to the *INO1* promoter and how this impacts the persistence and inheritance of memory will provide important insight into such epigenetic mechanisms of regulation.

Different molecular mechanisms regulate the initiation of transcriptional memory for different genes. Genes that exhibit salt-stimulated memory do not require Sfl1, suggesting that Sfl1 does not play a universal role in transcriptional memory. Likewise, while *INO1* memory requires the nuclear pore protein Nup100, salt stress-induced memory requires a different nuclear pore protein (Nup42) and is independent of Nup100 (*Guan et al., 2012*; *Light et al., 2010*). Distinct regulators of transcriptional poising may reflect distinct rate-limiting steps in their induction.

Despite these distinct regulators, this work has defined a conserved, core mechanism that leads to transcriptional memory involving histone H3 methylation and PIC poising (*Figure 8*). The genes that exhibit transcriptional memory that have been characterized to-date show the same chromatin changes during memory: persistent dimethylation of H3K4 without persistent acetylation or trimethylation of H3K4 (*Gialitakis et al., 2010*; *Light et al., 2013*). Dimethylation of H3K4 has also been implicated in forms of epigenetic memory leading to proper germline development in *Caenorhabditis elegans* and *Drosophila*, maintenance of acquired thermotolerance in plants and the acquisition of T cell memory (*Bevington et al., 2016*; *Lämke et al., 2016*; *Schaner et al., 2003*). Set1/COMPASS is the sole H3K4 methyltransferase in yeast, but in metazoan organisms, this complex has diversified (*Briggs et al., 2001*; *Dehé and Géli, 2006*; *Krogan et al., 2002*; *Roguev et al., 2001*). *Drosophila* expresses three COMPASS-related complexes: dSet1/COMPASS, d-Trithorax and d-Trithorax-Related complex with similar subunit compositions and humans express six COMPASS-related complexes (*Ardehali et al., 2011*; *Hughes et al., 2004*; *Lee et al., 2007*; *Mohan et al., 2011*; *Petruk et al., 2001*; *Shilatifard, 2008*; *Wu et al., 2008*). Despite their similarity, these complexes show specialization in their functions, predominantly producing H3K4me1, H3K4me2 or H3K4me3 (*Dou et al., 2006*; *Herz et al., 2014*; *Hu et al., 2013a*; *2013b*; *Jiang et al., 2013*; *2011*; *Mohan et al., 2011*; *Shilatifard, 2012*; *Steward et al., 2006*; *Wang et al., 2009*; *Wu et al., 2008*). Here we show that yeast COMPASS exists in two alternative forms in vivo that can produce either H3K4me3 (complete COMPASS) or H3K4me2 (COMPASS lacking Spp1), supporting previous work showing that COMPASS lacking Spp1 produces H3K4me2 but not H3K4me3 in vitro and in vivo (*Morillon et al., 2005*; *Schneider et al., 2005*; *Takahashi et al., 2009*). Thus, whereas metazoan systems have evolved several dedicated enzymes that separately catalyze mono- di- or trimethylation of histone H3 lysine 4, yeast generates these enzymes through regulated complex remodeling (*Figure 8A*). It remains to be seen if COMPASS remodeling occurs during memory in metazoan cells, or if a different COMPASS-related complex serves this function.

H3K4 dimethylation in yeast recruits SET3C, which is essential for *INO1* and salt stress transcriptional memory (*Figure 8A*). Although Set3 is recruited by H3K4me2 under both activating and memory conditions, it is only required for RNAPII binding and persistent H3K4 dimethylation under memory conditions, suggesting that SET3C might play a different role during activation. It is unclear if the SET3C deacetylase activity per se is important for establishing a poised state. Although memory is associated with hypoacetylated histones, neither loss of Set3 nor mutation of the MRS leads to acetylation under these conditions. Our results suggest that SET3C either protects H3 lysine 4 from demethylases and/or promotes recruitment of remodeled COMPASS during memory (*Figure 8A*). Homologous proteins may play a similar role in mammals since Set3 is similar to MLL5 and SET3C is related to NCoR/SMRT.

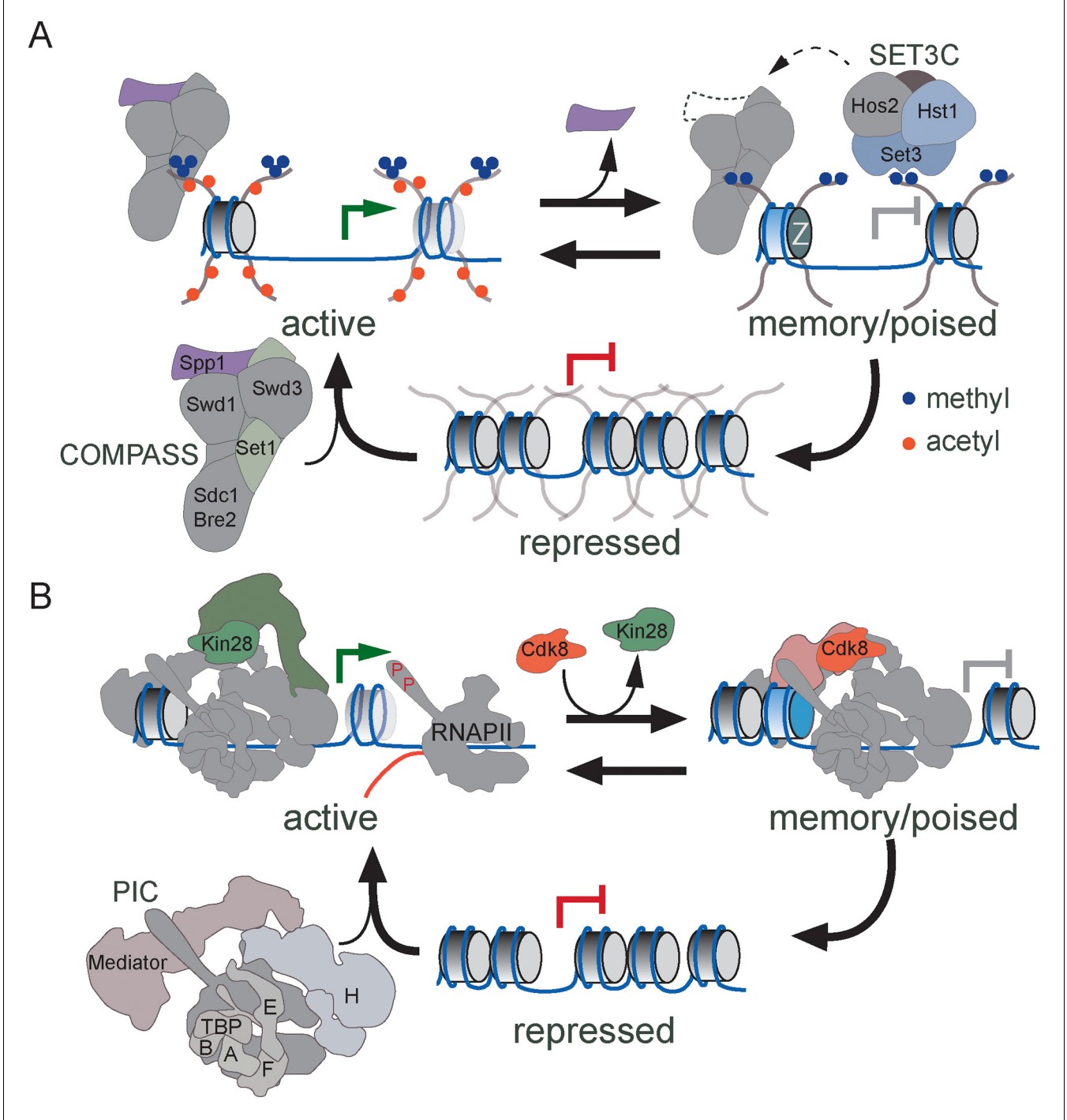

**Figure 8.** Models for transcriptional memory. (**A**) *Set1/COMPASS remodeling during INO1 transcriptional memory*. Nucleosomes associated with repressed *INO1* in the nucleoplasm are hypoacetylated and unmethylated. Active *INO1* is targeted to the nuclear periphery, nucleosomes are acetylated (orange circles) and H3K4 is trimethylated (blue circles) by COMPASS. During memory, *INO1* remains associated with the nuclear pore complex, acetylation is lost, H2A.Z is incorporated and H3K4 is dimethylated by a remodeled form of COMPASS lacking the Spp1 subunit (purple). H3K4me2 recruits Set3C, which promotes the persistence of H3K4me2 by feedback on COMPASS recruitment or remodeling. (**B**) *Cdk8+ Mediator promotes transcriptional poising*. Upon activation, Cdk8- Mediator and the PIC bind to the *INO1* promoter. TFIIK (Kin28/Cdk7) phosphorylates Serine 5 on the carboxy terminal domain of RNAPII to initiate transcription. During memory, Kin28 is lost and Cdk8+ Mediator is recruited. Cdk8+ Mediator promotes PIC recruitment but initiation is blocked by the absence of Kin28, poising the promoter for future activation.

The ultimate output of memory is the binding of the poised PIC, which presumably bypasses the rate-limiting step in transcriptional reactivation (*Figure 8B*). The recruitment of poised RNAPII requires H3K4 dimethylation, suggesting that the chromatin changes play an important role in PIC assembly. Assembly of the poised PIC during memory also requires TBP, Mediator and TFIIH, but is independent of Kin28/TFIIK (Cdk7). This is consistent with our observations that Kin28 is not bound to the *INO1* promoter during memory and that RNAPII is unphosphorylated on serine 5 of the CTD (*Light et al., 2010*; *2013*). Kin28 phosphorylation of the RNAPII CTD disrupts the interaction of Mediator with the PIC, allowing promoter escape (*Jeronimo and Robert, 2014*; *Wong et al., 2014*). Therefore, preinitiation poising may be achieved by recruitment of Mediator and PIC in the absence of Kin28.

PIC recruitment during *INO1* memory requires Cdk8$^+$ Mediator (*Figure 8B*); inactivation of Cdk8 disrupted RNAPII binding to the *INO1* promoter during memory, resulting in slower reactivation. Cdk8 has also been proposed to enhance transcription by directly phosphorylating transcription factors (*Bancerek et al., 2013*; *Nelson et al., 2003*) and RNAPII (*Knuesel and Taatjes, 2011*) and to regulate transcription elongation and pausing (*Galbraith et al., 2010*; *2013*). As a kinase, it is possible that Cdk8 has multiple, distinct roles (*Nemet et al., 2014*). However, for the genes that exhibit memory in yeast and humans, Cdk8 binding correlates with memory, when these genes are not transcribed. Furthermore, the role of Cdk8 in *INO1* memory is specific: inactivation of Cdk8 altered RNAPII recruitment to the *INO1* promoter under memory conditions and reduced the rate of reactivation, but had no effect on the RNAPII recruitment to the active gene or on the rate of activation. Reminiscent of the effect of Cdk8$^+$ Mediator on TFIIH recruitment to the poised human *RARB* promoter (*Pavri et al., 2005*), Cdk8$^+$ Mediator occupancy is mutually exclusive with Kin28/TFIIK during *INO1* memory. Thus, we propose that Cdk8$^+$ Mediator promotes transcriptional poising by facilitating recruitment of unphosphorylated RNAPII and, possibly, by regulating Kin28/TFIIK association with the PIC (*Figure 8B*; *Jeronimo and Robert, 2014*; *Wong et al., 2014*).

Genes that exhibit transcriptional memory are regulated by different mechanisms depending on the previous experiences of the cell. During the first experience of the stimulus, these genes are induced through regulated recruitment of RNAPII. After previous expression, these genes remain associated with poised RNAPII and their reactivation is stimulated by regulated initiation, a mechanism requiring Cdk8$^+$ Mediator. Thus, epigenetic transcriptional memory is a regulatory strategy that impinges upon a more broadly utilized poising mechanism through the action of nuclear pore proteins, remodeled COMPASS and SET3C. The function of H3K4 dimethylation is likely upstream of RNAPII PIC recruitment. This raises the possibility that some genes might be constitutively poised, independent of the memory-specific regulatory factors. Such genes should be bound to Cdk8$^+$ Mediator and be rapidly induced, but might not be dimethylated on H3K4 or associated with nuclear pore proteins.

## Materials and methods

### Chemicals and reagents

Unless noted otherwise, chemicals were from Sigma Aldrich, media components were from Sunrise Science and Genesee Scientific, oligonucleotides were from Integrated DNA Technologies, and restriction enzymes were from New England Biolabs. Secondary antibodies, Protein G dynabeads, and Pan-Mouse IgG Dynabeads were from Invitrogen. Other antibodies were from Abcam (anti-H3K4me2: ab32356, RRID:AB_732924; anti-GFP: ab290, RRID:AB_303395; anti-acetyl H3: ab47915, RRID:AB_873860), Millipore (anti-acetyl-H4 06–866, RRID:AB_310270), Thermo (anti-CDK8 PA1-21780, RRID:AB_2291488) or Covance (anti-RNAPII, 8WG16, RRID:AB_10063549). The anti-H3K4me2 antibody for the ChIP Seq was a generated in the Shilatifard Laboratory as described (*Thornton et al., 2014*).

### Yeast strains and cell lines

Yeast strains used in this study are listed in *Table 1*. GFP-tagged strains are from the genome-wide GFP strain collection from Open Biosystems (RRID:SCR_000808). The histone H3K4A and H3K4R mutants (*Dai et al., 2008*) were obtained from GE Lifesciences/Dharmacon (YSC5106). Genotypes are described using gene names from the Saccharomyces Genome Database (RRID:SCR_004694). HeLa

Cell Biology | Genes and Chromosomes

**Table 1.** Yeast strains.

| Name | Genotype | Figures | References |
|---|---|---|---|
| CRY1 | MATa ade2-1 can1-100 his3-11,15 leu2-3,112 trp1-1 ura3-1 | 1C-F 2A,B,D 4C-F, 7A and B | *Brickner & Walter, 2004* |
| ADY06 | MATα ade2-1 can1-100 his3-11, 15, leu2-3,112 trp1-1 ura3-1 set3Δ::KanMX | 4E and F 7C and D | *Light et al., 2013* |
| ADY20 | MATa ade2-1, can1-100, TFB1-GFP-FRB:HIS5$^+$ leu2-3,112 trp1-1 ura3-1 tor1-1 fpr1Δ::NAT RPL13A-2xFKBP12::TRP1 | 5G-I | This study |
| ADY21 | MATa ade2-1, can1-100, SPT15-GFP-FRB:HIS5$^+$ leu2-3,112 trp1-1 ura3-1 tor1-1 fpr1Δ::NAT RPL13A-2xFKBP12::TRP1 | 5A-C | This study |
| ADY22 | MATa ade2-1, can1-100, MED1-GFP-FRB:HIS5$^+$ leu2-3,112 trp1-1 ura3-1 tor1-1 fpr1Δ::NAT RPL13A-2xFKBP12::TRP1 | 5D-F | This study |
| ADY23 | MATa ade2-1, can1-100, SWD1-GFP-FRB:HIS5$^+$ leu2-3,112 trp1-1 ura3-1 tor1-1 fpr1Δ::NAT RPL13A-2xFKBP12::TRP1 | 2D-F, 3A | This study |
| ADY24 | MATa ade2-1, can1-100, SET3-GFP-FRB:HIS5$^+$ HOS2-GFP-FRB: KanMX, leu2-3,112 trp1-1 ura3-1 tor1-1 fpr1Δ::NAT RPL13A-2xFKBP12::TRP1 | 4A, G-I | This study |
| ADY31 | MATa ade2-1 can1-100, sfl1Δ::HIS3, leu2-3,112 trp1-1 ura3-1 | 1C 1D 1E 1F 2A | This study |
| WLY154 | MATa ade2-1 can1-100, his3-11,15, leu2-3,112 trp1-1 ura3-1 INO1-mrsmut | 1A 2A | *Light et al., 2013* |
| ADY32 | MATa ade2-1 can1-100 SFL1-FRB-GFP:HIS5$^+$, leu2-3,112 trp1-1 ura3-1 | 1A | This study |
| ADY33 | MATa ade2-1 can1-100 SFL1-FRB-GFP:HIS5$^+$ leu2-3,112 trp1-1 ura3-1 INO1-mrsmut | 1A | This study |
| JMY047 | MATa ade2-1 can1-100 his3-11,112 trp1-1 LacO:INO1:URA3 PHO88-mCherry:SpHis5, LEU2:LacI-GFP | 1B | This study |
| JMY049 | MATa ade2-1 can1-100 his3-11,112 trp1-1 LacO:INO1:URA3 PHO88-mCherry:SpHis5 LEU2:LacI-GFP sfl1Δ:: KanMX | 1B | This study |
| CEY272 | ade2-1 can1-100 his3-11,15 leu2-3,112 trp1-1 ura3-1 LEU2:pER05 HIS3:LacI-GFP URA3:LexA BS [pADH-LexA] | 1B | *Randise-Hinchliff et al., 2016* |
| CEY277 | ade2-1 can1-100 his3-11,15 leu2-3,112 trp1-1 ura3-1 LEU2:pER05 HIS3:LacI-GFP URA3:LexA BS [pADH-LexA-SFL1] | 1B | This study |
| WLY155 | MATa ade2-1 can1-100 his3-11,15 leu2-3,112 trp1-1 ura3-1 HIS3:pAFS144 TRP1:pRS304-Sec63-Myc INO1:p6LacO128-INO1 set1Δ::His5$^+$ | 2B | *Light et al., 2013* |
| ADY41 | MATa his3Δ200 leu2Δ0 lys2Δ0 trp1Δ63 ura3Δ0 met15Δ0 can1::MFA1pr-HIS3 hht1-hhf1::NatMX4 hht2-hhf2::[HHTS-HHFS]*-URA3 HHT-K4R | 2C and D | Dharmacon J. *Dai et al., 2008* |
| PJD47 | MATa his3Δ200 leu2Δ0 lys2Δ0 trp1Δ63 ura3Δ0 met15Δ0 can1::MFA1pr-HIS3 hht1-hhf1::NatMX4 hht2-hhf2::[HHTS-HHFS]*-URA3 wildtype HHT | 2C and D | Dharmacon J. *Dai et al., 2008* |
| ADY42 | MATa his3Δ200 leu2Δ0 lys2Δ0 trp1Δ63 ura3Δ0 met15Δ0 can1::MFA1pr-HIS3 hht1-hhf1::NatMX4 hht2-hhf2::[HHTS-HHFS]*-URA3 HHT-K4A | 2C and D | Dharmacon J. *Dai et al., 2008* |
| ADY34 | MATa ade2-1, can1-100, SPP1-GFP-FRB:HIS5$^+$ leu2-3,112 trp1-1 ura3-1 | 2E, 3A-C | This study |
| ADY35 | MATa ade2-1 can1-100, SET3-FRB-GFP:HIS5$^+$ leu2-3,112 trp1-1 ura3-1 | 4B | This study |
| ADY36 | MATa ade2-1 can1-100, SET3W140A-FRB-GFP:HIS5$^+$ leu2-3,112 trp1-1 ura3-1 | 4B-D | This study |
| ADY37 | MATa ade2-1 can1-100 SET3-FRB-GFP:HIS5$^+$ leu2-3,112 trp1-1 ura3-1 slf1Δ::KanMX | 4B | This study |
| ADY38 | MATa ade2-1 can1-100 KIN28-FRB:His5$^+$ leu2-3,112 trp1-1 ura3-1 tor1-1 fpr1Δ::NAT RPL13A-2xFKBP12::TRP1 | 4J 4K 4L | This study |
| ADY39 | MATa ade2-1 can1-100 SSN3-GFP-FRB:His5$^+$ leu2-3,112 trp1-1 ura3-1 tor1-1 fpr1Δ::NAT RPL13A-2xFKBP12::TRP1 | 6B-E | This study |
| HHY168 | MATα ade2-1 can1-100 his3-11,15 leu2-3,112 trp1-1 ura3-1 tor1-1 fpr1Δ::NAT RPL13A-2xFKBP12::TRP1 | 2Sup.1E | *Haruki et al., 2008* |
| Bre2-GFP | MATa his3Δ leu2ΔI met150ΔI ura3Δ0 BRE2-GFP:His5+ | 3A | Open Biosystems Ghaemmaghami et al. |
| Sdc1-GFP | MATa his3Δ leu2ΔI met150ΔI ura3Δ0 SDC1-GFP:His5+ | 3A | Open Biosystems Ghaemmaghami et al. |
| Med1-GFP | MATa his3Δ leu2ΔI met150ΔI ura3Δ0 MED1-GFP:His5+ | 6A | Open Biosystems Ghaemmaghami et al. |

*Table 1 continued on next page*

*Table 1 continued*

| Name | Genotype | Figures | References |
|------|----------|---------|------------|
| Med13-GFP | MATa his3Δ leu2ΔI met150ΔI ura3Δ0 MED13-GFP:His5+ | 6A | Open Biosystems Ghaemmaghami et al. |
| Ssn8-GFP | MATa his3Δ leu2ΔI met150ΔI ura3Δ0 SSN8-GFP:His5+ | 6A | Open Biosystems Ghaemmaghami et al. |

S3 cell line was used (RRID:CVCL_0058). The integration of LexA-LacO sequences at *URA3* for the LexA and Sfl1-LexA experiments in *Figure 1B* was performed as shown in and as described previously (*Ahmed et al., 2010*). The *mrs* mutation (or, as a control, wild-type *INO1*) was introduced into the endogenous *INO1* promoter via homologous recombination (*Ahmed et al., 2010*; *Light et al., 2010*). The *set3*-W140A point mutant was introduced into the *SET3* gene via homologous recombination as follows: *SET3* codon 140 was replaced with the *URA-SUP-o* double selection cassette by homologous recombination and selection for Ura+ Ade+ (*Randise-Hinchliff et al., 2016*). The W140A PCR product was then introduced in place of the *URA-SUP-o* cassette by homologous recombination and selection on 5-FOA and screening for pink colonies (*Randise-Hinchliff et al., 2016*).

All FRB Conditional strains were confirmed by microscopy for GFP florescence and PCR. Exponentially growing cultures were treated with a final concentration of 1 µg/ml rapamycin as previously described in *Haruki et al. (2008)* and performed lived imaging.

## Chromatin localization assay

Chromatin localization was performed as described (*Egecioglu et al., 2014*), using confocal microscopes in the Northwestern University Biological Imaging Facility.

## Western blot

Cell were lysed in 8M Urea 50 mM Hepes pH 7.5 by vortexing with glass beads for 4 min at 4°C. Pelleted cells harvested, and protein concentration was quantified using BCA assay (Pierce). 30–35 µg of lysate was separated on a 10% NuPage Bis-Tris gel in MES buffer (Invitrogen), transferred to nitrocellulose, and incubated overnight with antibodies against Tubulin, H3K4me, H3K4me2 and H3K4me3 in TBST+5% skim milk at 4°C. Blots were then washed twice with TBS, incubated with secondary antibody conjugated to HRP, and exposed to Enhanced Chemiluminescence reagents (Pierce) and imaged using a UVP BiospectrumAC Imaging System or Film.

## Chromatin immunoprecipitation

Yeast and HeLa ChIP was performed as described (*Egecioglu et al., 2014*) using primers in *Table 2*. Recovery of DNA was analyzed with primers described in *Table 2* and detailed in the legends of each figure. Anti-H3K4me2, -H3K4me3, -GFP -acetyl-H3, and anti-CDK8 were recovered with Rabbit-IgG Dynabeads and anti-RNAPII was recovered with Mouse-IgG Dynabeads. HeLa cells were grown to ~50% confluence, treated with 50 ng/mL of IFN-γ in DMEM supplemented with calf serum and antibiotics for 24 hr, washed extensively with PBS, trypsinized and ¼ of the plate was seeded to plates at appropriate densities that would lead to the same confluence when the cells were harvested 48 hr later.

## Reverse transcriptase real-time quantitative PCR

For experiments in which mRNA levels were quantified, RT-qPCR was performed as described (*Brickner et al., 2007*). Error bars represent the SEM of three biological replicates.

## ChIP-Seq

Yeast cells were fixed with 1% formaldehyde for 15 min, quenched with 150 mM glycine, washed with PBS and spun down. 400 µl of a wet cell pellet was suspended with 500 µl of FA lysis buffer (50 mM HEPES pH7.5, 140 mM NaCl, 1 mM EDTA, 1% Triton X-100, 0.1% sodium deoxycholate) supplemented protease inhibitor cocktail (Sigma, P8215). Cell suspension was mixed with 500 µl

**Table 2.** Oligonucleotides

| Primers Name | Sequence |
| --- | --- |
| INO1 Promoter FW | TCATCCTTCTTTCCCAGAATATTG |
| INO1 Promoter RV | CTCAAATTAACATTGCCGCC |
| INO1 CDS1 FW | TAGTTACCGACAAGTGCACGTACAA |
| INO1 CDS1 RV | TAGTCTTGAACAGTGGGCGTTACAT |
| INO1 CDS2 FW | GCGGAGGGGAATGACGTTTATG |
| INO1 CDS2 RV | CATATTCGAGAACTTGACTTCTCTGC |
| INO1 CDS3 FW | ACGCATCAGACGCGATATCCAG |
| INO1 CDS3 RV | CTGCAAGAGGTTTTCCATGGTGTC |
| ACT1CDS FW | GGTTATTGATAACGGTTCTGGTATG |
| ACT1CDS RV | ATGATACCTTGGTGTCTTGGTCTAC |
| PRM1 CDS FW | TAACAAGATTTGTCATCCAGCCTGC |
| PRM1 CDS RV | CCTCCTATACAAAATGGCCAATATG |
| GAL Promoter FW | CCCCACAAACCTTCAAATTAACG |
| GAL Promoter RV | CGCTTCGCTGATTAATTACCC |
| HSP31 promoter FW: | GAATTAACGTTACTCATTCCTAGCC |
| HSP31 promoter RV | TTTAAAGGGTAACGGAAACCGGAAG |
| HSP31 CDS FW: | GTTGGGATGAGCATTCCTTAGCC |
| HSP31 CDS RV: | ATAGTCAAATAAGGTACCGTGGCC |
| PGM2 promoter FW: | GGAACTTACGTGAAAGGGGACG |
| PGM2 promoter RV: | CCCACATTGTTCGGGCGGC |
| PGM2 CDS FW: | TGCCACTCTTGTTGTCGGTGGTG |
| PGM2 CDS RV: | GGTTCTCATGATGTGAGAAGCGGC |
| USV1 promoter FW: | AGTCTTCCGTATATAACAATCTCAATCC |
| USV1 promoter RV: | GTTAATGAAGCTGTTGCAAAATACTGC |
| USV1 CDS FW: | CTAGAGCGGAACATCTTGCACGTC |
| USV1 CDS RV: | GCTGGTGCGAGCTGGTAGAATGG |
| PMT5 promoter FW: | TCGCTCAAATAAGTATGATCTGCAAG |
| PMT5 promoter RV: | ACTACGCTTCTGTTCCTTTTCTATTG |
| PMT5 CDS FW: | CTGCCATCGTAAGGCTACACAATATC |
| PMT5 CDS RV | GAGGACACGGTTGCATATAGCATTG |
| GLC3 promoter FW: | ATATTACGGCATCATCTTTCCCCG |
| GLC3 promoter RV: | GGAAAATGGAAAGCCTTCCTTGC |
| GLC3 CDS FW: | TCATGCTACGCCTGATGGTTCG |
| GLC3 CDS RV: | CTCCCACTAGAAATGCACGTTCC |
| YGP1 promoter FW: | CTCTATTGCATCTTCAAACTCCGAAG |
| YGP1 promoter RV: | CAAGCTTTTTATATTTCAGAGATGATGG |
| YGP1 CDS FW: | GCCTGGAATGGGTCTAACTCTAGC |
| YGP1 CDS RV: | GGTGTAGTTTGTGTGGGTCAAAGAAC |
| HLA-DRA Pro For | GATTTGTTGTTGTTGTTGTCCTGTTTG |
| HLA-Dra Pro rev | GCAAATCAATTACTCTTTGGCCAATCAG |
| HLA-Dra CD For | GAAAGCAGTCATCTTCAGCGTT |
| HLA-DRA CD Rev | AGAGGCATTGGCATGGTGATAAT |
| CIITA Pro For | GTTCCCCCAACAGACTTTCTG |
| CIITA Pro Rev | AGGTGGCCCCAAGCGGTCAG |

*Table 2 continued on next page*

*Table 2 continued*

| Primers Name | Sequence |
| --- | --- |
| CITIA CD For | CACAGCCACAGCCCTACTTT |
| CIITA CD Rev | CCGACATAGAGTCCCGTGA |
| HLA-DPB1 Pro For | GGGCCAGCAGAATATTTGAGATCACC |
| HLA-DPB1 Pro Rev | GAGTCATTGCTCACTAGGCAGAAAGTTAG |
| HLA-DPB1 CD For | TCCAGCCTAGGGTGAATGTTTCCC |
| HLA-DPB1 CD Rev | TGGTGGACACGACCCCAGCTGTTTCCTCCTG |
| HLA-DQB1 Pro For | GGCACTGGATTCAGAACCTTCACAAA |
| HLA-DQB1 Pro Rev | CTGTGGATGTTTCCATGCGTGGTAGGATTGG |
| HLA-DQB1 CD For | CCCACAGTGACCATCTCCCCATCCAGGAC |
| HLA-DQB1 CD Rev | GGGGTGGACACAACGCCAGCTGTCTCCTCC |
| OAS2 Pro For | CAGTAAACCTTGCTGCAAGGGGCGGGGAAG |
| OAS2 Pro Rev | CCGGGACAGGGAAACAAAACTAACTTAAGC |
| OAS2 CD For | GGCTCCTATGGACGGAAAACAGTC |
| OAS2 CD Rev | CAACCACTTCGTGAACAGACAGAACTTC |
| URA3 FW | GACTCACTATAGGGCGAATTGGAGC |
| URA3 RV | GCCAAGCTCGGAATTAACCCTCAC |
| SUC2 Prom FW | CCTAAGGGCTCTATAGTAAACCATTTG |
| SUC2 Prom RV | GCACAAGAACAAGAGAATGTTTTGAAG |

0.5 mm glass beads from Biospec (11079105) in a 1.5 ml tube and lysed by vortexing in a TOMY multichannel mixer at 4°C for 1 hr to lyse cells. The sample was centrifuged at 1000x *g* at 4°C for 10 min. The pellet was suspended in 1 ml of FA lysis buffer + protease inhibitor cocktail and loaded into a 1 ml milliTUBE AFA Fiber from Covaris (520130). Chromatin shearing is performed with Covaris E220 focused ultrasonicator (Peak Incident Power: 280 W, Duty Factor; 20%, Cycles per Burst; 200, Time; 12 min). Sheared DNA sizes within 200 to 500 bp were confirmed by electrophoresis on a 2% agarose gel following protease K treatment at 65°C for 2 hr and DNA purification with QIAquick Spin Columns (QIAGEN). Sonicated chromatin was centrifuged in 1.5 ml tubes at maximal speed, 4°C for 15 min. The protein concentration of the cleared chromatin was quantified by Bradford method. 800 µl of 1.5 mg/ml chromatin in FA lysis buffer + protease inhibitor cocktail was mixed with 20 µl anti-H3K4me2 or H3K4me3 antibodies and incubated with gentle shaking at 4°C overnight. 100 µl of Protein A/G PLUS-agarose beads from Santa Cruz Biotech (sc-2003) equilibrated with FA lysis buffer, was added to the chromatin-antibody complex and incubated with gentle shaking at 4°C overnight. The beads are washed with FA lysis buffer once, FA lysis buffer supplemented with 1 M NaCl twice, FA-W3 buffer (10 mM TrisHCl pH8.0, 250 mM LiCl, 1 mM EDTA, 0.5% NP40, 0.5% sodium deoxycholate) once, and TE buffer once. Both the immunoprecipitated chromatin on the beads and non-immunoprecipitated chromatin for the input DNA were treated with 40 µg of protease K in 300 µl PK buffer (100 mM TrisHCl pH7.5, 150 mM NaCl, 12.5 mM EDTA, 1% SDS) at 42°C for 3 hr, followed by incubation at 65°C for 6 hr. The immunoprecipitated DNA and the input DNA were purified with QIAquick Spin Columns, and used for library construction with Illumina's TruSeq DNA Library preparation kit. DNA libraries were validated with a 2100 Bioanalyzer (Agilent Technologies) and sequenced with NextSeq 500 System (Illumina) using default Illumina standards for base calling and read filtering (*Keogh and Buratowski, 2004*).

## ChIP-Seq analysis

ChIP Seq Normalization: For the total number of aligned reads ($s_j$) from sample $j$, $y_{RAW}^{ij}$ is the raw reads coverage score at the $i^{th}$ position of $j^{th}$ sample. The normalized coverage score is calculated as follows:

$$y^{ij} = y^{ij}_{RAW} \times \frac{max_j(s_j)}{s_j}.$$

This global normalization method is effective and gave consistent baselines of reads coverage score across different samples throughout the genome for both di- and tri- methylation. We also generated a corresponding input control for each sample, which was used to locally normalize the reads. The results from the global normalization and two-step procedure (global + local) are extremely similar because the reads coverage score from the input samples is very similar across different samples throughout the genome. Therefore, we only presented the results based on the global normalization. *Figure 4—source data 1* includes five different comparisons between samples, as described in the legend.

## Acknowledgements

The authors thank Will Light, Richard Gaber, Danny Reinberg (New York University, HHMI) and members of the Brickner laboratory for helpful comments and suggestions on the manuscript and Professor Curt Horvath for help with HeLa cell experiments. This work was funded in part by the Chicago Biomedical Consortium with support from the Searle Funds at The Chicago Community Trust and by NIH grant R01 GM080484 (JHB). AD was supported by NIH T32 GM008061, JM & RC were supported by Northwestern Undergraduate Research Fellowships and Program in Biological Sciences Summer Grants and RC was also supported by a Krieghbaum Award. JHB is the Soretta and Henry Shapiro Research Professor in Molecular Biology.

## Additional information

### Competing interests

AS: Reviewing editor, *eLife*. The other authors declare that no competing interests exist.

### Funding

| Funder | Grant reference number | Author |
| --- | --- | --- |
| National Institute of General Medical Sciences | GM 080484 | Agustina D'Urso<br>Jessica Marone<br>Robert Coukos<br>Carlo Randise-Hinchliff<br>Jason H Brickner |
| Chicago Biomedical Consortium | Catalyst Award | Jason H Brickner |

The funders had no role in study design, data collection and interpretation, or the decision to submit the work for publication.

### Author contributions

AD, Conception and design, Acquisition of data, Analysis and interpretation of data, Drafting or revising the article; Y-hT, Acquisition of data, Analysis and interpretation of data, Drafting or revising the article, Contributed unpublished essential data or reagents; BX, J-PW, Analysis and interpretation of data, Drafting or revising the article; JM, Acquisition of data, Analysis and interpretation of data; RC, CR-H, Acquisition of data, Analysis and interpretation of data, Drafting or revising the article; AS, Analysis and interpretation of data, Drafting or revising the article, Contributed unpublished essential data or reagents; JHB, Conception and design, Analysis and interpretation of data, Drafting or revising the article

### Author ORCIDs

Jason H Brickner, http://orcid.org/0000-0001-8019-3743

# Additional files

## Major datasets

The following dataset was generated:

| Author(s) | Year | Dataset title | Dataset URL | Database, license, and accessibility information |
|---|---|---|---|---|
| D'Urso A, Takahashi Y, Marone J, Coukos R, Xiong B, Randise-Hinchliff C, Wang J, Shilatifard A, Brickner J | 2016 | Data from: COMPASS and Mediator are repurposed to promote epigenetic transcriptional memory | http://dx.doi.org/10.5061/dryad.93fv2 | Available at Dryad Digital Repository under a CC0 Public Domain Dedication |

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
