## [Decision Letter]

Thank you for submitting your article "COMPASS and Mediator are repurposed to promote epigenetic transcriptional memory" for consideration by *eLife*. Your article has been reviewed by three peer reviewers, and the evaluation has been overseen by Reviewing Editor Alan Hinnebusch and Kevin Struhl as the Senior Editor. The reviewers have opted to remain anonymous.

The reviewers have discussed the reviews with one another and the Reviewing Editor has drafted this decision to help you prepare a revised submission.

Summary:

Previous studies from Brickner has contributed to define molecular events associated with what they define as the transcriptional memory, eg, the facilitated activation of a gene that has undergone a cycle of activation/repression. Here, they further address how this memory is established. More precisely, they investigate the chromatin changes from an active state to a repressed state. They characterized a poised state which is supposed to be an intermediate state associated to the transcriptional memory. They show that the transition to active from repressed state (defined as memory condition) correlates with the binding of the transcriptional repressor SfI1 at the *INO1* promoter. This binding is dependent on the presence of a Memory Recruitment Sequence at the *INO1* promoter and correlates with high levels of H3K4me2 and PolII, two hallmarks of the poised state. Moreover, SfI1 is required to retain *INO1* at the nuclear envelope in the memory condition. Next, they show that PolII recruitment is lost in the memory condition when H3K4 methylation is abolishes. They further document this latter point by anchoring away the COMPASS Cps50 (Swd1) subunit which is essential for the assembly of the complex and therefore for the three H3K4 methylation states. By depleting Cps50 from the nucleus, they observed a rapid decrease of H3K4me2 but a slow decrease of PolII occupancy. In parallel of these observations, they found that the COMPASS subunit Cps40 (Spp1) which is only required for H3K4me3 is not recruited to *INO1* in memory conditions in contrast to other subunits (Cps50 (Swd1), Cps60 (Bre2), and Cps25 (Sdc1)). Consistent with previous studies from Buratowski lab, they show persistent H3K4me2 is correlated with the recruitment of Set3 to *INO1* in memory conditions in a way that depend on the integrity of the Set3 PHD domain and on SfI1. They show that the presence of Set3 allows H3K4me2 to be maintained in the 5' end and 3' end of *INO1* in memory conditions. Next they show that nucleus depletion of Spt15, Med1, and Tbf1 prevents PolII to be maintained at the *INO1* promoter in memory conditions while depletion of Kin28 prevents the binding of PolII under activation but not under memory conditions. Similarly, during memory removal of Kin28 does not affect H3K4me2. Along the same line, they show that the mediator subunit Ssn3 (Cdk8) binds to *INO1* promoter under memory and that its conditional depletion from the nucleus leads to the loss of PolII from the *INO1* promoter. Finally, they show that four genes whose expression is induced by H202 and that have been identified as exhibiting a salt stress memory share features of the *INO1* memory, eg, Set3 recruitment and Set3-dependent PolII occupancy under memory.

Essential revisions:

The following summarizes the important additional experiments and revision of text required by the referees, described more fully in the following complete reviews submitted by each of the three referees.

Reviewer #1:

The paper would benefit from a better description of the experimental strategies. Also, the authors should consider the suggested interpretation that the absence of Kin28 in the poised PIC could contribute to holding RNAPII at the promoter in memory conditions.

Reviewer #2:

– Figure 1: Show that changes in Sfl1 expression cannot account for its increased occupancy at *INO1* specifically in memory conditions; and show that Sfl1 binding at an inositol-independent promoter is constitutive (although showing this second point would likely make the first one superfluous.)

– Figure 1 much better explanation of the expectations of this assay for genes that either do, or do not, show localization to the nuclear periphery when they are actively transcribed is required here.

– From 1C and 1D: It is confusing to refer to Sfl1 as a repressor protein when it is required here for rapid reactivation. The several known roles of Sfl1 should be explained in the Introduction and later in the text, just refer to "Sfl1" rather than "Sfl1 repressor". It is also important to resolve discrepancies between the current results and those reported in Brickner et al. 2007 regarding the kinetics of *INO1* induction following different periods of repression in the memory state.

– Figure 2: The rationale for measuring H3K4 methylation only in the promoter region of *INO1* must be defended, or better yet, extended to include the CDS, possibly by using your genome-wide data. As discussed below, this criticism is underscored by the data shown later in Figure 4, indicating that most of the H3K4me2 signal is found in the CDS. It is also important to cite the previous literature relevant to these results and explain whether your data are providing novel insights or only confirming previous findings about di- and trimethylation of H3K4.

– Figure 3: In line with Reviewer #3's comments below, the expected loss of mono, di, and trimethylation of H3K4 in bulk chromatin following anchor away of Cps50 should be confirmed. Also, indicate that the *INO1* promoter is being analyzed in the figure legend.

– Figure 4: As noted above, it is necessary to resolve the important discrepancy between the ChIP-seq data in panel F showing a reduction in the H3K4me2 signal in the memory condition versus activating conditions in the CDS and even in the promoter, versus the previous conventional ChIP experiments which indicated no reduction in the promoter. There is also appreciable H3K4me2 present in repressing conditions in the ChIP-seq data but not in the previous conventional ChIP assays, which also requires explanation. The differences might arise from normalization differences, but the Materials and methods section lacks information about normalization, which must be rectified. Considering that Set3C is a deacetylase, it also seems important to determine the effect of eliminating Set3 on histone acetylation in the *set3* mutant, at least using conventional ChIP in the promoter, as the loss of acetylation in the memory state (Figure 2) could be dependent on Set3C and could contribute to the poised status of Pol II.

– Figure 5–Figure 6: The authors should more thoroughly discuss alternative models, e.g. whether or not they can distinguish the assembly of a poised PIC from a progressive disassembly of an active PolII that can recruit or stabilize other factors.

Reviewer #3:

1) It would be valuable to begin Results by describing the kinetics of *INO1* induction in the different states to lay out the memory phenomenon in terms of transcription levels. As noted above, resolving discrepancies in the kinetics between these data and those in the 2007 Brickner et al. paper is required.

2) The authors should put their findings into the context of previously published results on the poised state in other organisms, and indicate the novelty of their results.

5) It should be shown that the *mrs1* mutation eliminates H3K4me2 in the memory state.

6) It is important to establish the functionality of the CPS50-FRB and Cps50-FRB-GFP fusions by Western analysis of bulk H3K4me1, H3K4me2, H3K4me3.

7) It is necessary to cite the previous demonstrations that rapamycin has no effect in the strains engineered for anchor away.

8) As noted above, the expected loss of mono, di, and trimethylation of H3K4 following anchor away of Cps50 should be verified.

10) As the key conclusion of the paper, that COMPASS is repurposed during memory, is based on a single ChIP experiment in Figure 3 time course analysis of Cps40 occupancy would bolster this conclusion and also provide valuable information about when Cps40 is lost from the promoter during the establishment of the memory state.

11) As above, a more complete analysis of methylation that includes the CDS may be required at least for certain key experiments.

12) Stipulate the region of *INO1* being analyzed

13) As noted above, some measurement of histone deacetylation in WT versus set3 mutants is required.

15) In Figure 6, indicate the oligos used for the ChIP.

17) More complete citation of the relevant literature is required

18) COMPASS subunits should be labelled by their current yeast names.

---

## [Author Response]

*Essential revisions:*

*The following summarizes the important additional experiments and revision of text required by the referees, described more fully in the following complete reviews submitted by each of the three referees.*

*Reviewer #1:*

*The paper would benefit from a better description of the experimental strategies. Also, the authors should consider the suggested interpretation that the absence of Kin28 in the poised PIC could contribute to holding RNAPII at the promoter in memory conditions.*

We have added additional explanation of the chromatin localization assay and the ChIP seq data analysis.

We agree that the recruitment of Kin28 likely plays a critical role in transcriptional poising and we have rewritten sections of the Results and Discussion sections to strengthen that point.

*Reviewer #2:*

*– Figure 1: Show that changes in Sfl1 expression cannot account for its increased occupancy at INO1 specifically in memory conditions; and show that Sfl1 binding at an inositol-independent promoter is constitutive (although showing this second point would likely make the first one superfluous.)*

We have done both of these things. We now show that the nuclear localization and Sfl1 levels do not increase during memory using confocal microscopy of Sfl1-GFP. Also, we show that, whereas binding of Sfl1 to the *INO1* promoter occurs only during memory, binding to either another promoter (*SUC2*) or to an ectopic copy of the MRS element is unregulated. This suggests that Sfl1 binding to the *INO1* promoter is regulated by its context, either by neighboring transcription factors or by local changes in chromatin structure that are memory- specific.

*– Figure 1 much better explanation of the expectations of this assay for genes that either do, or do not, show localization to the nuclear periphery when they are actively transcribed is required here.*

We have augmented our description of the chromatin localization assay to include the following passage:

“Using confocal microscopy, the position of *INO1* was scored for colocalization with the nuclear envelope, visualized using a Pho88-mCherry fusion protein (Figure 1). Because the shell constituting the outer 25-30% of the nuclear volume (i.e. the 200nm closest to the nuclear envelope) is unresolvable from the nuclear envelope by light microscopy, a randomly localized spot within the nucleus is expected to colocalize with the nuclear envelope in ~30% of the cells (blue hatched line, Figure 1; Brickner & Walter, 2004).”

*– From 1C and 1D: It is confusing to refer to Sfl1 as a repressor protein when it is required here for rapid reactivation. The several known roles of Sfl1 should be explained in the Introduction and later in the text, just refer to "Sfl1" rather than "Sfl1 repressor". It is also important to resolve discrepancies between the current results and those reported in Brickner et al. 2007 regarding the kinetics of INO1 induction following different periods of repression in the memory state.*

We now refer to Sfl1 as a “transcription factor” rather than as a repressor.

We do not think there are any significant discrepancies between the current work and that reported in Brickner et al., 2007. The reviewer may be referring to the precise kinetics of *INO1* induction as measured by RT qPCR, which can vary depending on the strains used, stocks of inositol, the primers or the qPCR reagents. However, these effects have no bearing on the conclusion of either study.

One thing that is counter-intuitive about *INO1* memory is that the rate of reactivation is similar to, or even slightly slower than, the rate of activation. This is particularly true in certain strains like BY4741 (see Brickner et al., 2007, Figure 4). The experiments conducted here, in Light et al., 2010 and in Light et al., 2013 used W303-based strains, for which this difference is less clear. The reason that reactivation is not faster than activation is due to the fact that the Ino1 enzyme is long-lived and cells cannot sense inositol starvation as quickly during reactivation as during activation. This creates a refractory period during reactivation that is not observed during activation, delaying the onset of transcription. However, once they sense starvation, the rate of reactivation is strongly dependent on transcriptional memory. To strengthen our proposal that *INO1* is primed for faster activation, we now include kinetic analysis of RNAPII binding to both the promoter and the coding sequence during either activation or reactivation in wild type and *sfl1∆* mutant strains (Figure 1). This experiment shows clearly that memory promotes association of active RNAPII with the *INO1* gene during reactivation, enhancing the rate of reactivation.

Furthermore, we have also analyzed transcriptional memory for salt-induced genes in yeast and IFN-γ-induced genes in human cells, neither of which have the issues associated with *INO1*. For both sets of genes, reactivation is either much faster than activation or much stronger (Light et al., 2013; Figure 7 in the current manuscript). This makes us confident that, despite the complexities of inositol biology, *INO1* is an excellent model for a conserved and broadly utilized mechanism of transcriptional poising.

– Figure 2: The rationale for measuring H3K4 methylation only in the promoter region of INO1 must be defended, or better yet, extended to include the CDS, possibly by using your genome-wide data. As discussed below, this criticism is underscored by the data shown later in Figure 4, indicating that most of the H3K4me2 signal is found in the CDS. It is also important to cite the previous literature relevant to these results and explain whether your data are providing novel insights or only confirming previous findings about di- and trimethylation of H3K4.

We now include qPCR results showing that H3K4me2 is associated with the promoter and the 5’ end of the *INO1* gene. We also include a brief discussion of the possible sources of the differences between the qPCR experiments and the ChIP-seq experiment. Finally, we include additional, seminal citations regarding the methylation status of active and repressed genes and clarify what is new in this work.

– Figure 3: In line with Reviewer #3's comments below, the expected loss of mono, di, and trimethylation of H3K4 in bulk chromatin following anchor away of Cps50 should be confirmed. Also, indicate that the INO1 promoter is being analyzed in the figure legend.

We have added immunoblots examining the loss of H3K4me3 and H3K4me1 after addition of rapamycin to the Swd1-FRB-GFP strain and to the parent strain lacking Swd1-FRB-GFP.

In all figure legends, we indicate the position of the *INO1* primers used for qPCR.

*– Figure 4: As noted above, it is necessary to resolve the important discrepancy between the ChIP-seq data in panel F showing a reduction in the H3K4me2 signal in the memory condition versus activating conditions in the CDS and even in the promoter, versus the previous conventional ChIP experiments which indicated no reduction in the promoter. There is also appreciable H3K4me2 present in repressing conditions in the ChIP-seq data but not in the previous conventional ChIP assays, which also requires explanation. The differences might arise from normalization differences, but the Materials and methods section lacks information about normalization, which must be rectified.*

We agree that this is a likely explanation. As mentioned above, qPCR analysis of ChIP gives a slightly different picture than ChIP-seq. We include a brief discussion of this issue in the Results section and we have edited the description of ChIP-seq analysis in the Materials and methods section.

Considering that Set3C is a deacetylase, it also seems important to determine the effect of eliminating Set3 on histone acetylation in the set3 mutant, at least using conventional ChIP in the promoter, as the loss of acetylation in the memory state (Figure 2) could be dependent on Set3C and could contribute to the poised status of Pol II.

We have now analyzed the H3 and H4 acetylation in both the *mrs* mutant and *set3∆* strains. We observed no increase in the acetylation during memory in these mutant strains.

*– Figure 5–Figure 6: The authors should more thoroughly discuss alternative models, e.g. whether or not they can distinguish the assembly of a poised PIC from a progressive disassembly of an active PolII that can recruit or stabilize other factors.*

We now include a more thorough discussion of alternative models for regulating transcriptional poising through Kin28 recruitment (see above). However, we think that the progressive disassembly model is not supported by the data. *INO1* is repressed extremely rapidly (Brickner et al., 2007). The RNAPII associated with *INO1* memory persists for 4 cell divisions, suggesting that it cannot represent the persistent binding of RNAPII to the originally transcribed gene (which constitutes only 6.25% of the loci after 4 generations). Memory lasts for up to 7 generations in HeLa cells (our unpublished data). Thus, RNAPII must be continuously recruited over this time period. Recruitment requires factors upstream of Kin28, but not Kin28, which is incompatible with the model that it is slow disassembly of active PIC (which requires Kin28). Finally, the recruitment of RNAPII during memory requires factors that are bound only after repression and that have no effect on the active promoter (i.e. Sfl1, Cdk8, Nup100).

*Reviewer #3:*

1) It would be valuable to begin Results by describing the kinetics of INO1 induction in the different states to lay out the memory phenomenon in terms of transcription levels. As noted above, resolving discrepancies in the kinetics between these data and those in the 2007 Brickner et al. paper is required.

We do not think there are any significant discrepancies between the current work and that reported in Brickner et al., 2007. The reviewer may be referring to the precise kinetics of *INO1* induction as measured by RT qPCR, which can vary depending on the strains used, stocks of inositol, the primers or the qPCR reagents. However, these effects have no bearing on the conclusion of either study.

One thing that is counter-intuitive about *INO1* memory is that the rate of reactivation is similar to, or even slightly slower than, the rate of activation. This is particularly true in certain strains like BY4741 (see Brickner et al., 2007, Figure 4). The experiments conducted here, in Light et al., 2010 and in Light et al., 2013 used W303-based strains, for which this difference is less clear. The reason that reactivation is not faster than activation is due to the fact that the Ino1 enzyme is long-lived and cells cannot sense inositol starvation as quickly during reactivation as during activation. This creates a refractory period during reactivation that is not observed during activation, delaying the onset of transcription. However, once they sense starvation, the rate of reactivation is strongly dependent on transcriptional memory. To strengthen our proposal that *INO1* is primed for faster activation, we now include kinetic analysis of RNAPII binding to both the promoter and the coding sequence during either activation or reactivation in wild type and *sfl1∆* mutant strains (Figure 1). This experiment shows clearly that memory promotes association of active RNAPII with the *INO1* gene during reactivation, enhancing the rate of reactivation.

Furthermore, we have also analyzed transcriptional memory for salt-induced genes in yeast and IFN-γ-induced genes in human cells, neither of which have the issues associated with *INO1*.

For both sets of genes, reactivation is either much faster than activation or much stronger (Light et al., 2013; Figure 7 in the current manuscript). This makes us confident that, despite the complexities of inositol biology, *INO1* is an excellent model for a conserved and broadly utilized mechanism of transcriptional poising.

2) The authors should put their findings into the context of previously published results on the poised state in other organisms, and indicate the novelty of their results.

We have now discussed and cited a number of additional studies with relevance to our current manuscript and clarified the novelty of our contribution.

*5) It should be shown that the mrs1 mutation eliminates H3K4me2 in the memory state.*

This is included in Figure 2.

6) It is important to establish the functionality of the CPS50-FRB and Cps50-FRB-GFP fusions by Western analysis of bulk H3K4me1, H3K4me2, H3K4me3.

We have added immunoblots examining the loss of H3K4me3 and H3K4me1 after addition of rapamycin to the Swd1-FRB-GFP strain and to the parent strain lacking Swd1-FRB-GFP.

7) It is necessary to cite the previous demonstrations that rapamycin has no effect in the strains engineered for anchor away.

We now make reference to the original Haruki et al., 2008 paper, which analyzed the effect of rapamycin on growth of a strain lacking the FRB-tagged protein. Also, we performed immunoblots against H3K4me1 and H3K4me3 from this strain and ChIP against H3K4me2 and RNAPII under repressing, activating and memory conditions. We observed no phenotype upon addition of rapamycin.

8) As noted above, the expected loss of mono, di, and trimethylation of H3K4 following anchor away of Cps50 should be verified.

See response to point 6, above.

10) As the key conclusion of the paper, that COMPASS is repurposed during memory, is based on a single ChIP experiment in Figure 3 time course analysis of Cps40 occupancy would bolster this conclusion and also provide valuable information about when Cps40 is lost from the promoter during the establishment of the memory state.

We agree that the ChIP experiment alone could be strengthened by the suggested time course experiment. Therefore, we have examined the Spp1 (Cps40) binding upon repression and upon reactivation (Figure 3). These results completely support our initial result and provide a clear view of the remodeling of COMPASS during memory and upon reactivation.

11) As above, a more complete analysis of methylation that includes the CDS may be required at least for certain key experiments.

We have included new qPCR data in Figure 2 showing that *INO1* is marked with H3K4me2 over the 5’end and promoter during memory.

12) Stipulate the region of INO1 being analyzed

We have added descriptions of the location of the primers used for each experiment to the legends.

*13) As noted above, some measurement of histone deacetylation in WT versus set3 mutants is required.*

We have tested this hypothesis and added these results to the manuscript. Loss of Set3 has no effect on the acetylation of the *INO1* promoter under any condition and therefore, it is unclear whether the deacetylase activity plays a critical role in memory. We have included this conclusion in the Results and Discussion sections.

15) In Figure 6, indicate the oligos used for the ChIP.

We have added descriptions of the location of the primers used for each experiment to the legends.

17) More complete citation of the relevant literature is required

We have now discussed and cited a number of additional studies with relevance to our current manuscript and clarified the novelty of our contribution.

18) COMPASS subunits should be labelled by their current yeast names.

We have changed the names of the COMPASS subunits in the text and the Figures to the appropriate names.